# DefensiveKV: Taming the Fragility of KV Cache Eviction in LLM Inference

**Yuan Feng[1,3,†], Haoyu Guo[2,3,†], Junlin Lv[1,3], S. Kevin Zhou[2,3], Xike Xie[2,3,*],**

[1]School of Computer Science, University of Science and Technology of China
[2]School of Biomedical Engineering, USTC
[3]Data Darkness Lab, MIRACLE Center, Suzhou Institute for Advanced Research

## Abstract

Large language models have revolutionized natural language processing, yet their deployment remains hampered by the substantial memory and runtime overhead of the transformer's Key-Value cache. To mitigate this, recent methods employ a scoring-aggregation framework to evict unimportant cache entries, based on the "stability assumption"—that a fixed subset of entries remains consistently important during generation. However, prior work has largely focused on refining importance indicators for scoring, while defaulting to mean aggregation due to a faithful trust in the stability assumption. In this work, we argue that this underlying assumption is inherently fragile, making mean aggregation highly vulnerable in extreme cases. To counter this, we propose a simple yet elegant defensive aggregation strategy: a two-step, linear-time approach that controls worst-case risk, thereby defending against extreme cases with negligible computational overhead. Embodying this strategy, we propose a novel cache eviction method, DefensiveKV and its extension, Layer-DefensiveKV, which incorporates layer-wise budget allocation. Across seven task domains (18 datasets), our methods reduce generation quality loss by 2.3× and 4.3× respectively, versus the strongest baseline under a 20% cache size. These results set new performance benchmarks and pioneer a promising direction for optimizing cache eviction against underlying fragility through worst-case risk management. Our code is available at `https://github.com/FFY0/DefensiveKV`.

## 1 Introduction

Transformer-based Large Language Models (LLMs) have enabled a wide range of applications (Yi et al., 2024; Gu, 2023). Due to their autoregressive nature, LLMs maintain a Key-Value (KV) cache to store intermediate representations of previously tokens, which supports efficient computation of future generation. However, as the input sequence length increases, the KV cache grows linearly, leading to substantial overhead. For example, a 70B-parameter model with a batch size of 8 and a sequence length of 128k may require up to 330GB of memory just for caching. This poses significant challenges regarding storage expenses and I/O bottlenecks for LLM deployment (Devoto et al., 2025).

Early solutions like StreamingLLM (Xiao et al., 2023) reduce cache size by keeping only recent cache entries, but this sacrifices long-range context. More recent solutions (Zhang et al., 2024b; Liu et al., 2024c; Li et al., 2024; Feng et al., 2025) on selective cache eviction operate under a key underlying assumption that *a fixed subset of cache entries remain consistently important and contributes to future generation*. Thus, by retaining the selected subset, the full KV cache can be approximated with a much smaller memory footprint. Building on this, existing methods typically follow a two-step scoring-aggregation framework: In the scoring step, different historical token queries are used to observe multiple importance scores for each past KV cache entry. In the aggregation step, these multiple observed scores for each cache entry are aggregated—typically by averaging—to estimate its expected significance and guide the eviction strategy.

---

[†]Equal Contribution:    {yfung,haoyuguo}@mail.ustc.edu.cn
[*]Corresponding Author:    xkxie@ustc.edu.cn

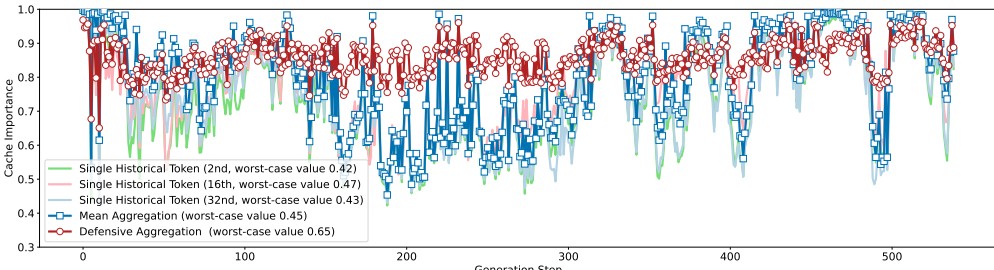

Figure 1: Defensive aggregation demonstrates robustness against fragile stability assumption (Llama-3.1-8B, 50% cache size, layer 14, summary task). Appendix L provides additional visualizations.

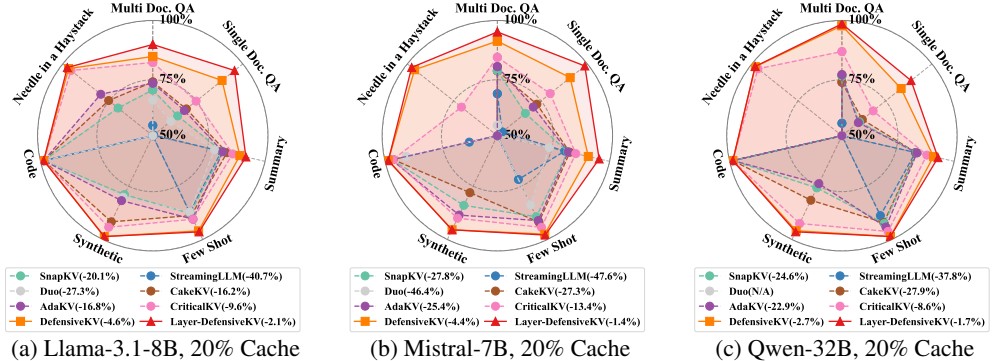

(a) Llama-3.1-8B, 20% Cache     (b) Mistral-7B, 20% Cache     (c) Qwen-32B, 20% Cache

Figure 2: DefensiveKV and Layer-DefensiveKV achieve significantly lower losses of generation quality compared to all baselines across various domains and models.

Following this two-step framework, previous research has primarily focused on improving the scoring step by exploring various importance indicators. Early studies often relied solely on naive attention weights (Zhang et al., 2024b; Liu et al., 2024c; Oren et al., 2024). SnapKV (Li et al., 2024) improved by introducing a pooling mechanism, while more recent work like CriticalKV (Feng et al., 2025) employed the norm of projected value states to offer a more principled measure of importance. However, the second step—aggregating these importance scores—remains largely underexplored. Most existing methods default to a simple averaging strategy. While this may seem reasonable, averaging is only effective if the underlying assumption holds that importance are stable-when it does, averaging helps reduce observation noise and capture the consistent significance of cache entries.

> *There raises a critical question: if the stability assumption proves unreliable, is averaging still the best aggregation strategy, or might better alternatives exist?*

In this work, we show that even when the assumption generally holds, it remains inherently fragile, as importance scores can shift abruptly during generation. As demonstrated in Figure 1, performing cache compression based on the observed importance score of a single historical token often yields promising results, with most steps retaining over 0.8 correlation with full-cache importance. However, in certain intervals (e.g., steps 150–230), the stability assumption breaks down—consequently, results based on single historical token fail, leading to sharp drops, with some outliers falling as low as 0.5. In these cases, the current standard practice of mean aggregation, simply averaging these single token predictions, inevitably results in similar outlier performance.

> *This reflects a classic pitfall, a flaw directly analogous to a foundational lesson from finance: strategies that optimize only for the average case (expected returns) are fundamentally flawed because they ignore the risk of rare but extreme negative cases (worst-case risks).*

Inspired by this insight, we abandon average-case optimization in favor of a worst-case risk management framework for KV cache eviction, which we term defensive aggregation. Our strategy is actualized through an elegant two-step process: worst-case estimation and adaptive prior-risk correction. Remarkably, this approach requires only two linear-time operations, matching the computational efficiency of standard mean aggregation. As shown in Figure 1, Defensive Aggregation demonstrates clear superiority, boosting the worst-case retained importance to 0.65—a substantial improvement over both mean aggregation (0.45) and single-token baselines (0.42, 0.47, 0.43).

Building on the defensive aggregation strategy, we introduce DefensiveKV, a general cache eviction method, which we further develop into Layer-DefensiveKV by leveraging a popular layer-wise budget allocation strategy. Figure 2 summarizes that these two methods significantly outperform prior approaches across seven task domains, evaluated on 18 datasets from the LongBench and Needle-in-a-Haystack benchmarks. With a 20% cache budget, DefensiveKV and Layer-DefensiveKV incur generation quality losses of only 4.8% and 2.6%, respectively, representing 2.3× and 4.3× reductions versus the best baseline, CriticalKV (11.1%).

## 2 RELATED WORKS

KV cache eviction is crucial for efficient long-sequence inference in LLMs. Current approaches largely fall into two paradigms based on their primary objective. *Fixed-Pattern Framework*: This paradigm prioritizes operational efficiency, employing fixed rules for cache retention. Early methods like StreamingLLM (Xiao et al., 2023) retained only recent cache entries, which is highly efficient but often loses valuable long-range information. More recent work, such as LaCache (Shi et al., 2025), uses a ladder-shaped structure. These methods are advantageous in scenarios where compression occur frequently, such as at every decoding step, where their low computational overhead is a core benefit. However, this efficiency often comes at the cost of compression quality; for instance, StreamingLLM can incur over 40% loss with 20% cache size as summarized in Figure 2. *Scoring-Aggregation Framework*: The second and more dominant paradigm, which our work addresses, focuses on compression quality. These methods are better suited for common, less frequent compression scenarios. These methods, examplified by H2O (Zhang et al., 2024b) and Scissorhands (Liu et al., 2024c) introduced importance-based eviction, operating under the "cache importance stability" assumption—that a small set of entries remains consistently important. They follow a scoring-aggregation framework: they observe importance multiple times and then aggregate these scores, often by averaging, to decide on eviction Ren & Zhu (2024); Oren et al. (2024).

Within the dominant scoring-aggregation framework, prior research has almost exclusively focused on improving the importance observation (scoring) step. For example, SnapKV (Li et al., 2024) introduced pooling, and CriticalKV (Feng et al., 2025) proposed using projected value norms for a more principled score. However, the foundational stability assumption and the subsequent aggregation step have rarely been rigorously examined. This paper revisits and reveals the fragility of this assumption, further showing prevalent mean aggregation's vulnerability. Consequently, we are the first to underscore the necessity of risk-control defensive aggregation strategies to against fragile assumption. This pioneers a new research direction, entirely orthogonal to prior work focused on optimizing importance indicators. For demonstration, we build our DefensiveKV method upon CriticalKV, the current SOTA importance indicator.

Additionally, our contributions is also orthogonal to various KV cache budget allocation strategies, including intra-layer (e.g., AdaKV (Feng et al., 2024)), inter-layer (e.g., PyramidKV (Zhang et al., 2024a), LightTransfer (Zhang et al., 2025), CAKE (Qin et al., 2025)), and also offline training-based allocation (e.g., HeadKV (Fu et al., 2024), DuoAttention (Xiao et al., 2025)). These strategies focus on optimizing budget allocation for cache eviction methods, and are thus inherently orthogonal to our investigation. Direct comparison is not essential for validating our contributions. However, to demonstrate our principles' adaptability, we introduce Layer-DefensiveKV, a variant using layer-wise budget allocation for enhancement. Broader related methods like quantization, channel pruning, and sparse attention are discussed in Appendix F. Furthermore, we provide a case study in Appendix H on integrating our DefensiveKV with quantization, showing minimal loss even at 10% cache footprint.

## 3 METHODS

### 3.1 PRELIMINARY

LLM generation consists of two stages, *prefilling* and *decoding*. During prefilling, the KV states for all input tokens are computed and cached as: $K = HW_K$, $V = HW_V$, where $H \in \mathbb{R}^{n \times d}$ denotes the hidden states for $n$ tokens, and $W_K, W_V \in \mathbb{R}^{d \times d_h}$ are learned matrices. In decoding, the LLM takes the most recent token, computes its query vector $q_j = H_{j=-1,:}W_Q$, and retrieves information from the cached KV entries using attention to produce the output $o_j$ and predict the next token:

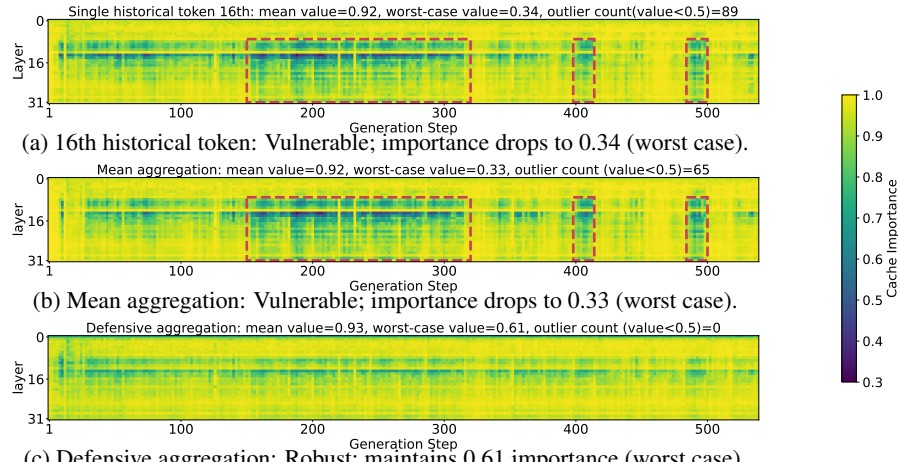

(a) 16th historical token: Vulnerable; importance drops to 0.34 (worst case).

(b) Mean aggregation: Vulnerable; importance drops to 0.33 (worst case).

(c) Defensive aggregation: Robust; maintains 0.61 importance (worst case).

Figure 3: Mean vs. Defensive Aggregation: Against Fragility in Importance Stability

$$o_j = A_j V W_O \text{ where } A_j = \text{softmax}\left(q_j K^\top / \sqrt{d_h}\right)$$

To reduce the memory overhead of maintaining the full KV cache, cache eviction methods have been developed. These methods largely operate under a stability assumption: *a fixed subset of KV cache entries, denoted as $(\hat{K}, \hat{V})$, retains consistent importance throughout generation.* Based on this assumption, the objective of cache eviction is to identify this crucial subset $(\hat{K}, \hat{V})$ using historical queries (i.e., tokens from earlier generation process), and use it to replace the full KV cache $(K, V)$ in subsequent steps. This process typically follows a two-step *scoring-aggregation* framework, where the importance of each KV entry is first estimated (or scored) and then aggregated:

1. **Scoring.** Given $m$ historical tokens, represented as queries $Q = [q_1, \ldots, q_m]$, each of the $n$ KV cache entries $K, V \in \mathbb{R}^{n \times d_h}$ is scored. This results in an importance matrix $I \in \mathbb{R}^{m \times n}$, where each element $I_{j,i}$ measures the relevance of the $i$-th KV cache entry $(k_i, v_i)$ for the $j$-th historical query $q_j$. In practice, the attention weight $A_{j,i}$ serves as a direct measure of importance $I_{j,i}$, given the attention mechanism's inherent weighted-sum formulation.

2. **Aggregation.** Subsequent aggregation step converts the observed importance matrix $I \in \mathbb{R}^{m \times n}$ into a vector $S \in \mathbb{R}^n$, where each $S_i$ represents the importance of the $i$-th KV cache entry. Existing works adopt mean aggregation, $S_i = \frac{1}{m} \sum_{j=1}^{m} I_{j,i}$, to highlight entries with consistently high importance, in line with the stability assumption.

While subsequent studies have refined the scoring step—SnapKV employs a pooling mechanism, and CriticalKV utilizes the norm of projected value states $v_i W_O$ for more principled scoring—the aggregation step has received little attention. This is largely because mean aggregation appears to align closely with underlying importance stability assumption. However, in this work, we show the fragility of that assumption and reveal the vulnerability of mean aggregation under this premise. This underscores the necessity of revisiting and improving the aggregation step.

### 3.2 Fragility of Stability Assumption and Vulnerability of Mean Aggregation

We examine this using Llama3.1-8B on the Government Report summarization task. Adopting the SOTA importance indicator, $I_{j,i} = A_{j,i} \times \text{norm}(v_i W_O)$, we observe each cache entry importance with 32 recent historical tokens $q_j$. We then simulate a 50% cache eviction using two different criteria. The first criterion uses importance scores from a single historical token observation, while the second uses scores averaged across all 32 historical tokens (mean aggregation). For each result, we track the proportion of total importance it retains during subsequent generation, relative to the full cache.

**Fragility of the Stability Assumption.** Figure 3a presents the results for the 16th historical token(see Appendix M for results from other tokens). The results reveal a general, yet fragile, stability. On average, the 50% retained cache subset accounts for 0.92 of the full cache's total importance during generation. However, this high average belies the underlying fragility: the retained importance can

Table 1: Defensive aggregation consistently improves the worst-case values across all task types.

| Task type
Dataset | Single-Doc. QA
NrtvQA | Multi-Doc. QA
HotpotQA | Summary
GovReport | Few-shot
TREC | Synthetic
PCount | Code
Lcc |
|---|---|---|---|---|---|---|
| Mean aggregation | 0.44 | 0.39 | 0.28 | 0.47 | 0.47 | 0.30 |
| Defensive aggregation | **0.62** | **0.60** | **0.52** | **0.61** | **0.61** | **0.50** |

---

**Algorithm 1** Defensive Aggregation

---

1: **Input:** Importance scores $I \in \mathbb{R}^{m \times n}$, where $I_{j,i}$ is the importance of entry $i$ based on historical token $j$
2: **Output:** Aggregated risk scores $\tilde{R} \in \mathbb{R}^n$
3: $\tilde{R}_i = \max_{1 \leq j \leq m} I_{j,i}$, $\forall i = 1, \ldots, n$          ▷ Worst-case Risk Estimation
4: $R_i = \max\left(\tilde{R}_i, \bar{R}\right)$ where $\bar{R} = \frac{1}{n} \sum_{i=1}^{n} \tilde{R}_i$, $\forall i = 1, \ldots, n$     ▷ Adaptive Prior-Risk Correction
5: **return** $R$

---

drop sharply, as seen in the interval between steps 150 and 320. In these moments of instability, the worst-case retained importance drops to as low as 0.34. Additionally, outliers—where the retained 50% of the cache captures less than half of the total importance (value $< 0.5$)—are frequent, occurring in 89 instances in this trial alone.

**Vulnerability of Mean Aggregation.** Current eviction methods commonly employ mean aggregation over multiple importance observations. The rationale is to obtain an expected importance to guide eviction. However, by failing to account for worst cases, this strategy becomes vulnerable precisely under the fragile assumption, leading to outlier performance similar to that of using a single, unreliable token. As shown in Figure 3b, significant drops persist at the same problematic steps observed in Figure 3a, reaching a worst-case importance value of 0.33 and resulting in 65 outlier instances. This outcome is predictable. The observation score based on single historical token is inherently blind to the fragility of stability assumption; thus it cannot hedge against the worst-case risk. While simple averaging acts as a form of "reconciliation" among these individual observations to produce a moderate result, it does not incorporate any mechanism to control for this underlying risk. Consequently, it cannot aggregate a prediction to consistently outperform every single-token observation. When most single token-based observations fail, the mean-aggregated result is inevitably dragged down with them, thus offering no meaningful improvement and producing similarly damaging outliers.

This underscores a critical point: rather than focusing solely on designing more accurate importance indicators, it is equally—if not more—important to develop new aggregation methods explicitly designed for worst-case risk control, which can provide reliable estimates even when most single-token observations fail.

3.3    DEFENSIVE AGGREGATION VIA WORST-CASE RISK CONTROL

Consider a cache entry may exhibits high importance in only a few single-token observations while remaining low in most others duo to fragile stability. Mean aggregation would not recognize this as important and would erroneously evict it. When this entry becomes crucial again in future generation, the prior eviction results in substantial importance loss. Thus, relying on mean aggregation fails to guard against these extreme cases. To address this, we introduce a novel *defensive aggregation*, a novel strategy that eschews simple averaging in favor of a worst-case risk control perspective as shown in Algorithm 1.

**Worst-case Risk Estimation.** From a risk-control perspective, the penalty for evicting a KV cache entry is equivalent to the importance score it would have possessed at future moment. The "worst-case risk", $R_i^*$, is therefore the peak importance score an entry could attain over the entire future generation process. If we denote the future generated sequence as $L$, then $R_i^* = \max_{t \in L} I_{t,i}$. As this future maximum is unknowable at eviction time, we instead approximate it as the maximum importance score observed across all $j$ historical tokens, e.g. $\tilde{R}_i = \max_{1 \leq j \leq m} I_{j,i}$, $\forall i = 1, \ldots, n$. [1]. This $O(n)$ procedure matches mean aggregation's runtime yet yields significantly better empirical performance, as it better captures the potential worst-case risk if the entry were removed.

---

[1] For Grouped-Query Attention, a cache entry's worst-case risk estimate is the maximum importance score observed over historical tokens across all heads sharing its KV group.

**Adaptive Prior-Risk Correction.** Although the above estimator takes the maximum over observed history, it could still underestimate worst-case risk because eviction methods typically restrict the observation window (e.g., 32 tokens) to limit overhead. [2] Such restricted observations could miss rare but critical risks. Inspired by Laplace smoothing in Bayesian estimation, we introduce an adaptive prior–risk correction. For each head, define a head-level prior risk $\bar{R} = \frac{1}{n}\sum_{i=1}^{n}\tilde{R}_i$, i.e., the average observed worst-case risk across entries for that head. If the observed risk $R_i$ falls below prior risk $\bar{R}$, we treat the shortfall as under-observation and substitute the prior: $R_i = \max\left(\tilde{R}_i, \bar{R}\right)$. Thereby, heads with higher overall risk receive larger priors, reducing reliance on limited historical observations. The effectiveness of correction and its adaptive design is validated in Section 4.4.

By defending against risks of the fragile stability assumption, our defensive aggregation substantially improves worst-case performance compared to mean aggregation. As shown in Figure 3c, it boosts the worst-case retained importance from 0.33 to 0.61 and completely eliminates the 65 outlier instances produced by mean aggregation. Table 1 further confirms this advantage is consistent across six datasets with different task types. Therefore, this simple two-operation method provides a crucial defense against the fragility of the importance stability assumption.

### 3.4 IMPLEMENTING DEFENSIVEKV EVICTION METHOD WITH DEFENSIVE AGGREGATION

Building upon our proposed defensive aggregation strategy, we introduce two novel cache eviction methods: *DefensiveKV* and *Layer-DefensiveKV*.

**DefensiveKV** serves as the foundational variant. It directly integrates defensive aggregation into the traditional cache eviction workflow by replacing the conventional mean aggregation. Despite its simplicity, this modification alone leads to substantial performance improvements.

**Layer-DefensiveKV** further refines this by incorporating a layer-wise budget allocation, inspired by existing strategies (Feng et al., 2024; Zhang et al., 2024a). It performs a joint selection of risky entries across layers, enabling more budget to be allocated to layers with more risky cache entries.

As shown in Algorithm 2, the overall process of DefensiveKV adheres to the established practices: preserving the KV cache entries of several recent historical tokens (Line 3) and then utilizing the query states of these tokens for importance measurement. The importance calculation begins with basic attention weights (Line 4) and incorporates further refinements—specifically pooling mechanisms from SnapKV (Line 5), and projected value norm scaling from CriticalKV (Line 7) [3].

The key innovation in DefensiveKV is the strategic replacement of conventional mean aggregation with our defensive aggregation (Line 6). This simple modification, requiring minimal changes, reduces over $2\times$ in generation quality loss. The extension to Layer-DefensiveKV is also straightforward. It incorporates two additional refinements: first, projected value norms are normalized layer-wise to address their variance across layers (Line 11); second, risky entries are selected jointly across all layers (Line 12). This leads to an even more impressive gain, with over $4\times$ reduction in quality loss.

## 4 EXPERIMENTS

### 4.1 EXPERIMENTAL SETTINGS

**Models.** We evaluate our approach on three open-source LLMs: Llama-3.1-8B-Instruct (lla, 2024; Touvron et al., 2023) and Qwen2.5-32B-Instruct (Team, 2024), supporting context lengths of up to 128K, and Mistral-7B-Instruct-v0.3 (Jiang et al., 2023), supporting up to 32K.

**Baselines.** We compare our method against six baselines. StreamingLLM (Xiao et al., 2023) is an early sliding window approach. SnapKV (Li et al., 2024), AdaKV (Feng et al., 2024), and CAKE (Qin et al., 2025) use attention weight-coupled pooling for importance indicators; CAKE also employs a cascaded architecture for layer-wise budget allocation. The SOTA CriticalKV (Feng et al., 2025) introduces a more accurate importance indicator. DuoAttention (Xiao et al., 2024b), a training-based

---

[2]Explicitly computing attention weights for all tokens is infeasible with FlashAttention optimization, and even storing all attention weights is prohibitively expensive (e.g., $\approx$64 GB for a 32k context in Llama-3.1-8B).

[3]Although our method is based on current SOTA practice, defensive aggregation is widely applicable to other eviction methods. Appendix J includes a case study applying it to different baseline methods for demonstration.

---

**Algorithm 2** (Layer)-DefensiveKV

---

1: **Input:** Cache Entries $K, V$, Parameter $W_O$, queries of $m$ recent historical tokens $Q = [q_1, ..., q_m]$
2: **Output:** Retained KV Cache $\hat{K}, \hat{V}$
3: Append the KV cache of recent historical tokens $K[-m:,:], V[-m:,:]$ to $\hat{K}, \hat{V}$.
4: $A \leftarrow \text{softmax}(QK^T/\sqrt{d})$
5: $A \in \mathbb{R}^{m \times n} \leftarrow \text{Pooling}(A, \text{dim} = -1)$    ▷ Refined with pooling by SnapKV (Li et al., 2024)
6: $R \in \mathbb{R}^n \leftarrow$ Defensive Aggregation Algorithm 1 $(A)$    ▷ Our modification
7: $R_i \leftarrow R_i \times \text{norm}(v_i W_O)$  $\forall v_i \in V, i = 1, \ldots, n$ ▷ Refined with norm by CriticalKV (Feng et al., 2025)
8: **if** without layer-wise budget allocation **then**    ▷ Leading to DefensiveKV
9:    Select the cache entries with top worst-case risk $R$ independently in each layer and append to $\hat{K}, \hat{V}$
10: **else**    ▷ Leading to Layer-DefensiveKV
11:    $R_i \leftarrow R_i / \sum_i \text{norm}(v_i W_O)$  $\forall v_i \in V, i = 1, \ldots, n$
12:    Select the cache entries with top worst-case risk $R$ jointly across all layers and append to $\hat{K}, \hat{V}$
13: **end if**
14: **return** $\hat{K}, \hat{V}$

---

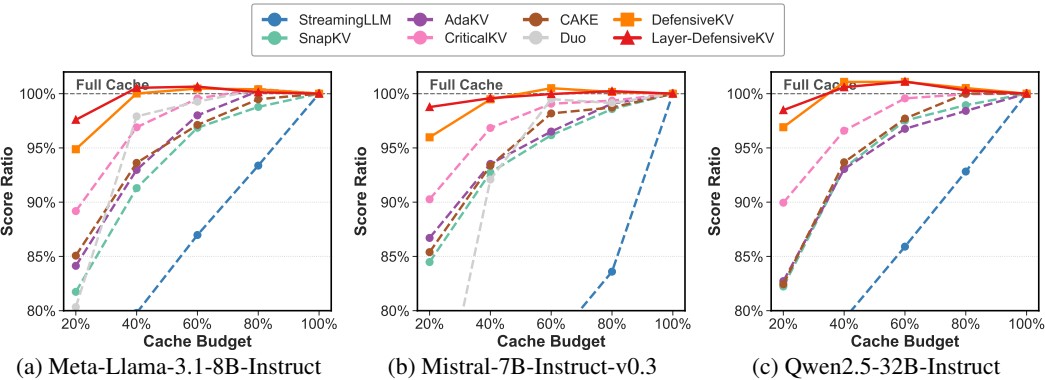

(a) Meta-Llama-3.1-8B-Instruct    (b) Mistral-7B-Instruct-v0.3    (c) Qwen2.5-32B-Instruct

Figure 4: Overview of averaged generation quality across 16 datasets on LongBench.

method, is included with official configurations for Llama-3.1-8B and Mistral-7B-v0.3, but marked N/A for Qwen2.5-32B due to unavailable configurations and high training costs. All methods use a historical window size of 32 and are accelerated with FlashAttention-2(Dao et al., 2022; Dao, 2023).

**Settings.** Following the settings in (Devoto et al., 2025; Feng et al., 2025), the context is compressed independently before question is introduced. This better simulates practical scenarios (e.g., multi-turn QA or prefixed contexts) where multiple questions often pertain to the same context, or the question is unavailable during context compression. Thus this setup is more challenging and better reflects the real-world performance of cache eviction methods. (Feng et al., 2025)

### 4.2 LONGBENCH EVALUATION

LongBench (Bai et al., 2024) serves as a comprehensive benchmark, featuring 16 datasets structured into six task domains: single-document QA, multi-document QA, summary, few-shot learning, synthetic tasks, and code completion. Detailed information for each dataset can be found in Appendix N

**Overall Analysis.** Figure 4 illustrates our methods' significant advantages in average quality loss across 16 datasets. As cache size drops from 100% to 40%, all baselines degrade noticeably, while our DefensiveKV and Layer-DefensiveKV remain nearly lossless. For instance, with a 40% cache on Llama-3.1-8B (Figure 4a), CriticalKV (best baseline) loses 3.1% quality, whereas our DefensiveKV shows no degradation, surpassing even the training-based DuoAttention (2.2% drop, despite offline training costs). At a smaller 20% cache, CriticalKV's loss is 10.6%, while DefensiveKV limits it to 5.1%(over 2x reduction). Our Layer-DefensiveKV further cuts this loss to 2.3%(over 4x reduction). Similar advantages hold for other models. For instance, on Mistral-7B with 20% cache, DefensiveKV and Layer-DefensiveKV achieve 4.0% and 1.3% loss, respectively, far below CriticalKV's 9.7%.

**Task Analysis.** Figure 5 displays Llama-3.1-8B average scores by task domain (Appendix A for more models). DefensiveKV and Layer-DefensiveKV consistently excel. While simpler task

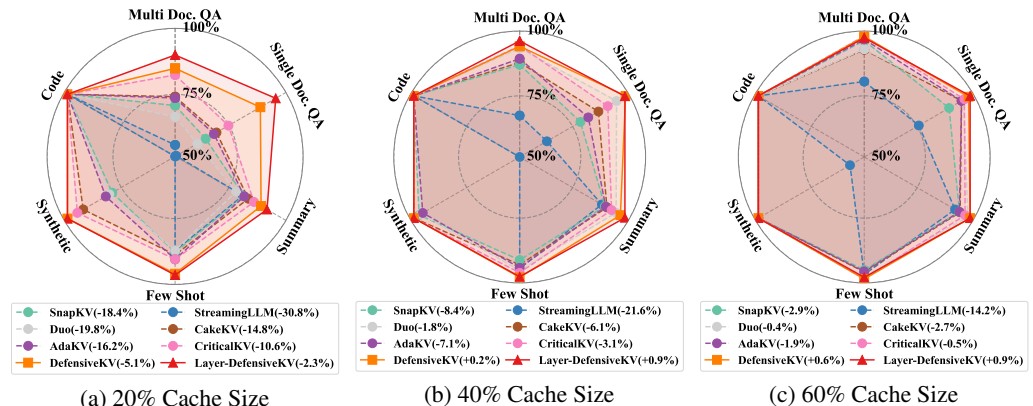

(a) 20% Cache Size     (b) 40% Cache Size     (c) 60% Cache Size

Figure 5: Analysis of the six task domains on LongBench for Meta-Llama-3.1-8B-Instruct.

Table 2: Detailed scores of 16 datasets on LongBench.

| Method | Single-Document QA | | | Multi-Document QA | | | Summarization | | | Few-shot Learning | | | Synthetic | | Code | | |
|---|---|---|---|---|---|---|---|---|---|---|---|---|---|---|---|---|---|
| | NrtvQA | Qasper | MF-en | Hotpot | 2WikiQA | Musique | GovRep | QMSum | MultiNews | TREC | TriviaQA | SAMSum | PCount | PR-en | Lcc | RB-P | Avg. |
| Llama-3.1-8B-Instruct, 20% Cache Size | | | | | | | | | | | | | | | | | |
| Full Cache | 29.55 | 44.68 | 55.82 | 57.59 | 48.89 | 32.61 | 34.40 | 25.51 | 26.83 | 73.00 | 92.36 | 43.27 | 7.38 | 99.50 | 63.44 | 52.36 | 49.20 |
| DuoAttention | 23.28 | 21.22 | 34.03 | 42.89 | 28.14 | 20.57 | 25.32 | 19.48 | 23.12 | 56.00 | 86.54 | 40.67 | 7.50 | 78.50 | 65.94 | 59.19 | 39.52 |
| StreamingLLM | 22.05 | 19.83 | 23.87 | 39.44 | 20.97 | 15.46 | 27.76 | 20.63 | 22.27 | 53.50 | 89.97 | 40.04 | 4.00 | 29.50 | 65.61 | 60.66 | 34.72 |
| SnapKV | 25.64 | 28.23 | 29.71 | 46.17 | 29.64 | 22.07 | 27.09 | 21.51 | 22.46 | 48.50 | 92.21 | 44.08 | 5.08 | 79.50 | 67.17 | 54.36 | 40.21 |
| CAKE | 26.29 | 30.54 | 33.28 | 46.03 | 32.08 | 24.73 | 27.77 | 22.16 | 22.91 | 51.50 | 91.86 | 43.56 | 6.50 | 92.50 | 65.46 | 52.50 | 41.85 |
| AdaKV | 27.07 | 28.69 | 32.85 | 49.64 | 30.89 | 21.57 | 26.70 | 21.85 | 22.67 | 55.50 | 91.30 | 43.89 | 7.30 | 80.50 | 66.44 | 55.43 | 41.39 |
| CriticalKV | 29.81 | 32.58 | 34.96 | 52.34 | 36.24 | 26.37 | 28.35 | 23.52 | 23.24 | 56.50 | 90.80 | 43.37 | 8.89 | 93.00 | 67.05 | 54.99 | 43.88 |
| DefensiveKV | 29.97 | 40.46 | 46.23 | 52.20 | 38.40 | 28.06 | 29.96 | 23.89 | 24.11 | 68.00 | 91.58 | 43.17 | 8.28 | 100.00 | 67.17 | 55.40 | 46.68 |
| Layer-DefensiveKV | 30.10 | 42.91 | 52.94 | 55.03 | 44.07 | 27.00 | 30.99 | 24.95 | 24.42 | 69.00 | 91.30 | 43.54 | 8.38 | 100.00 | 67.60 | 56.00 | 48.01 |
| Mistral-7B-Instruct-v0.3, 20% Cache Size | | | | | | | | | | | | | | | | | |
| Full Cache | 27.02 | 38.19 | 50.22 | 50.75 | 37.41 | 27.92 | 34.45 | 25.76 | 26.37 | 76.00 | 89.01 | 46.89 | 6.50 | 97.00 | 66.04 | 60.47 | 47.50 |
| DuoAttention | 11.91 | 13.58 | 29.88 | 31.73 | 22.43 | 9.18 | 23.96 | 17.25 | 22.67 | 49.50 | 86.08 | 43.08 | 2.67 | 18.00 | 59.89 | 56.23 | 31.13 |
| StreamingLLM | 18.30 | 16.38 | 26.26 | 38.78 | 25.99 | 15.06 | 28.00 | 20.73 | 21.32 | 30.50 | 80.88 | 40.57 | 3.00 | 28.00 | 32.62 | 46.91 | 29.58 |
| SnapKV | 21.91 | 23.69 | 30.59 | 43.71 | 28.28 | 19.81 | 27.91 | 21.15 | 22.15 | 55.00 | 89.41 | 46.67 | 5.00 | 82.50 | 64.30 | 59.97 | 40.13 |
| CAKE | 23.08 | 25.42 | 35.31 | 44.10 | 28.96 | 18.59 | 28.27 | 21.18 | 22.61 | 60.00 | 90.36 | 46.40 | 4.00 | 77.00 | 64.50 | 59.21 | 40.56 |
| AdaKV | 24.00 | 26.29 | 31.15 | 45.26 | 27.99 | 20.65 | 27.37 | 21.67 | 22.38 | 59.00 | 89.87 | 46.27 | 4.50 | 88.00 | 65.05 | 59.50 | 41.18 |
| CriticalKV | 24.14 | 29.56 | 38.91 | 45.42 | 32.08 | 21.26 | 28.59 | 22.71 | 23.11 | 65.50 | 90.13 | 46.65 | 4.11 | 90.00 | 65.42 | 58.43 | 42.88 |
| DefensiveKV | 21.05 | 34.67 | 50.05 | 48.76 | 32.27 | 26.09 | 31.95 | 23.39 | 24.05 | 72.50 | 90.11 | 46.80 | 3.53 | 96.50 | 65.78 | 61.93 | 45.59 |
| Layer-DefensiveKV | 27.31 | 39.41 | 49.70 | 49.89 | 37.82 | 24.16 | 33.13 | 25.08 | 25.49 | 74.50 | 89.61 | 46.25 | 3.06 | 97.00 | 66.99 | 61.21 | 46.91 |
| Qwen2.5-32B-Instruct, 20% Cache Size | | | | | | | | | | | | | | | | | |
| Full Cache | 30.88 | 46.13 | 52.87 | 63.59 | 59.75 | 38.78 | 32.59 | 24.35 | 24.95 | 72.00 | 88.26 | 47.05 | 12.50 | 100.00 | 49.64 | 34.24 | 48.60 |
| DuoAttention | N/A | N/A | N/A | N/A | N/A | N/A | N/A | N/A | N/A | N/A | N/A | N/A | N/A | N/A | N/A | N/A | N/A |
| StreamingLLM | 20.74 | 17.68 | 25.04 | 39.53 | 33.19 | 17.38 | 27.83 | 19.04 | 21.67 | 61.00 | 82.01 | 42.80 | 10.67 | 29.00 | 56.20 | 30.14 | 33.37 |
| SnapKV | 24.52 | 24.58 | 28.80 | 52.54 | 42.07 | 29.74 | 28.25 | 19.45 | 21.35 | 58.00 | 87.74 | 48.21 | 9.00 | 76.35 | 53.19 | 35.58 | 39.96 |
| CAKE | 22.25 | 26.49 | 31.33 | 49.20 | 42.48 | 28.11 | 27.86 | 18.92 | 22.02 | 58.00 | 87.77 | 47.22 | 11.00 | 81.25 | 51.82 | 35.25 | 40.06 |
| AdaKV | 25.49 | 22.51 | 29.13 | 54.11 | 41.24 | 29.98 | 28.01 | 19.33 | 21.81 | 61.50 | 88.09 | 48.02 | 9.00 | 74.00 | 53.95 | 35.14 | 40.20 |
| CriticalKV | 29.65 | 25.93 | 32.51 | 58.92 | 48.60 | 34.54 | 29.77 | 20.86 | 22.23 | 65.00 | 88.44 | 48.65 | 10.50 | 94.75 | 53.95 | 35.14 | 43.72 |
| DefensiveKV | 31.11 | 32.11 | 45.66 | 62.55 | 57.68 | 40.55 | 30.30 | 22.06 | 22.98 | 71.00 | 88.96 | 48.05 | 9.50 | 99.75 | 54.29 | 36.98 | 47.10 |
| Layer-DefensiveKV | 31.38 | 35.65 | 49.25 | 64.24 | 58.86 | 40.25 | 30.86 | 23.07 | 23.10 | 75.00 | 88.92 | 47.47 | 10.00 | 99.88 | 52.21 | 35.62 | 47.86 |

domains (Code, Synthetic) show high performance for most methods, challenging ones (Doc QA, Summarization) reveal significant baseline degradation under reduced cache. Our methods maintain their advantages. For instance, in Single-doc QA (20% cache), CriticalKV (strongest baseline) drops to 74.8% of full-cache quality; DefensiveKV achieves 89.6%, and Layer-DefensiveKV reaches 96.7%. Table 2 further reports detailed 20% cache scores (other results in Appendix K). On Llama-3.1-8B (20% cache), DefensiveKV beats CriticalKV on 13/16 datasets; Layer-DefensiveKV wins on 15/16. Such a significant performance advantage, rarely observed between other baselines, highlights the effectiveness of our "worst-case risk" perspective to against underlying fragility across diverse tasks.

## 4.3 NEEDLE-IN-A-HAYSTACK EVALUATION

In the Needle-in-a-Haystack test, the key sentence is placed in a long context to evaluate retrieval ability. Following Ruler (Hsieh et al., 2024), we test two representative cases with a 32K context length: (1) Single-retrieval: one needle is randomly inserted for retrieval. (2) Multi-retrieval: four needles are randomly inserted and all need to be retrieved. Further details, along with evaluations on more "needle-in-a-haystack-style" tasks from Ruler, are provided in Appendix I.

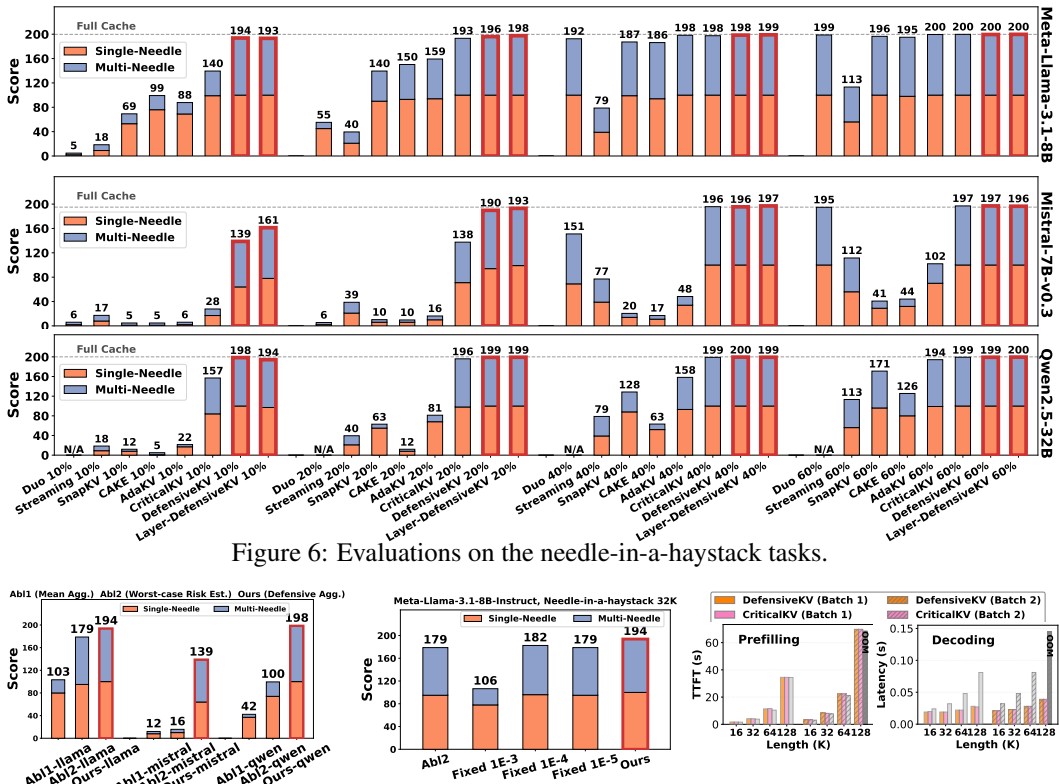

Figure 6: Evaluations on the needle-in-a-haystack tasks.

Figure 7: Ablation 10% cache.  Figure 8: Adaptive Correction  Figure 9: Efficiency(FlashAttn2)

As shown in Figure 6, our DefensiveKV and Layer-DefensiveKV achieve significantly higher scores across all settings. For instance, on long-context models like Llama-3.1-8B and Qwen2.5-32B (both supporting 128K context length), our methods maintain near-lossless, with scores 194 and 193 for Llama-3.1-8B at mere 10% cache size. In contrast, even the strongest baseline, CriticalKV, drops to 140 under the same conditions, while others fall below 100—demonstrating a substantial gap. On weaker long-context ability model, i.e., Mistral-7B (maximum context length 32K), all baselines suffer severe performance degradation. At a 10% cache size, most baselines score below 6, and CriticalKV only reaches 28. However, our DefensiveKV and Layer-DefensiveKV achieve scores of 139 and 161, over 5x and 5.8x improvements, respectively.

## 4.4 ABLATION STUDIES

**Component ablation.** To analyze the roles of the two key operations in our defensive aggregation—worst-case risk estimation and adaptive prior-risk correction—we conduct ablation studies based on our DefensiveKV method. First, we only remove adaptive prior-risk correction (denoted as Abl2). Then, we further ablate the worst-case risk estimation by replacing with common mean aggregation (denoted as Abl1). As shown in Figure 7, using only worst-case risk estimation (Abl2) already significantly outperforms that are with mean aggregation. For example, on Llama-3.1-8B, Abl2 improves the score from 103 (Abl1) to 179. Adding our adaptive prior-risk correction provides further gains, with our full DefensiveKV method reaching a score of 194. These results confirm that both operations contribute meaningfully to the overall performance.

**Adaptive correction analysis.** To validate the adaptive design of our prior-risk correction, we ablated it against fixed correction thresholds (1E-3, 1E-4, 1E-5). The results in Figure 8 show that fixed thresholds are ineffective. Most fail to outperform the no-correction baseline score of 179 (Abl2), with the 1E-4 case providing only a marginal gain to 182. Our adaptive correction, however, reaches a score of 194, confirming that tailoring the correction to each head's risk profile is crucial. Additionally, the hyperparameter-free nature of our adaptive design ensures consistently strong performance across two additional models (shown as Abl2 vs. ours in Figure 7).

Table 3: Ablation study on the historical window size.

| Method | Single-Document QA | | | Multi-Document QA | | | Summarization | | | Few-shot Learning | | | Synthetic | | Code | | |
|---|---|---|---|---|---|---|---|---|---|---|---|---|---|---|---|---|---|
| | NrtvQA | Qasper | MF-en | Hotpot | 2WikiQA | Musique | GovRep | QMSum | MultiNews | TREC | TriviaQA | SAMSum | PCount | PR-en | Lcc | RB-P | Avg. |
| Llama-3.1-8B-Instruct, 20% Cache Size | | | | | | | | | | | | | | | | | |
| Full Cache | 29.55 | 44.68 | 55.82 | 57.59 | 48.89 | 32.61 | 34.40 | 25.51 | 26.83 | 73.00 | 92.36 | 43.27 | 7.00 | 99.50 | 63.44 | 52.36 | 49.20 |
| CriticalKV | 29.81 | 32.58 | 34.96 | 52.34 | 36.24 | 26.37 | 28.35 | 23.52 | 23.24 | 57.00 | 90.80 | 43.37 | 8.89 | 93.00 | 67.05 | 54.99 | 43.88 |
| DefensiveKV (16) | 29.75 | 35.43 | 46.54 | 55.51 | 36.41 | 25.64 | 29.20 | 23.68 | 23.34 | 66.00 | 91.69 | 44.02 | 7.31 | 100.00 | 67.00 | 55.10 | 46.02 |
| DefensiveKV (32) | 29.97 | 40.46 | 46.23 | 52.20 | 38.40 | 28.06 | 29.96 | 23.89 | 24.11 | 68.00 | 91.58 | 43.17 | 8.28 | 100.00 | 67.17 | 55.40 | 46.68 |
| DefensiveKV (64) | 28.98 | 36.09 | 45.49 | 56.69 | 39.31 | 27.50 | 30.63 | 24.12 | 23.31 | 68.00 | 91.13 | 43.35 | 6.77 | 100.00 | 66.72 | 55.04 | 46.41 |
| Layer-DefensiveKV (16) | 29.64 | 41.23 | 49.21 | 54.59 | 41.31 | 29.05 | 30.19 | 24.22 | 24.25 | 68.50 | 91.86 | 44.04 | 9.00 | 99.50 | 68.82 | 55.00 | 47.53 |
| Layer-DefensiveKV (32) | 30.10 | 42.91 | 52.94 | 55.03 | 44.07 | 27.00 | 30.99 | 24.95 | 24.42 | 69.00 | 91.30 | 43.54 | 8.38 | 100.00 | 67.60 | 56.00 | 48.01 |
| Layer-DefensiveKV (64) | 30.51 | 43.40 | 51.42 | 55.36 | 40.03 | 26.23 | 31.60 | 24.70 | 24.29 | 70.00 | 91.61 | 43.21 | 6.92 | 100.00 | 68.69 | 56.46 | 47.75 |

**Window size robustness.** In our main experiments, we adopted the default historical window size of 32, following the setting of the strongest baseline, CriticalKV Feng et al. (2025). To analyze the impact of this hyperparameter, we conduct an ablation study on the Llama-3.1-8B-Instruct model (20% cache size) using historical window sizes of 16, 32, and 64. The results are presented in Table 3. Both DefensiveKV and Layer-DefensiveKV consistently and substantially outperform the CriticalKV baseline (43.88) across all tested window sizes. DefensiveKV achieves average scores ranging from 46.02 to 46.68, while Layer-DefensiveKV scores between 47.53 and 48.01. This study demonstrates that the effectiveness of our proposed methods is robust to variations in the historical window size.

## 4.5 EFFICIENCY TEST

We compare DefensiveKV and CriticalKV, which differ only in their aggregation mechanisms, to demonstrate that defensive aggregation introduces negligible computational overhead. Our experiments, conducted on an 80GB A100 GPU with Llama-3.1-8B (20% cache), show in Figure 9 that DefensiveKV and CriticalKV have nearly identical time-to-first-token (TTFT) and decoding latency. All KV cache eviction occurs during the prefilling stage and is included in TTFT, confirming that DefensiveKV adds negligible overhead. Additionally, cache eviction significantly reduces decoding latency versus Full Cache: e.g., for batch size 1 and a 128K context length, latency drops from 0.081s (Full Cache) to 0.028s with eviction-based methods (a 2.9x speedup). Furthermore, cache eviction allows larger batch sizes; for example, eviction methods can handle batch size of 2, while Full Cache results in out-of-memory errors, leading to a 4.2× decoding throughput boost. See Appendix G for memory usage details.

## 5 CONCLUSION

This work challenges the fragile stability assumption underlying existing KV cache eviction methods. We show that widely used mean aggregation strategies are highly vulnerable under the fragile stability, resulting in poor worst-case performance. To address this, we propose "defensive aggregation", a novel strategy explicitly designed from a "worst-case risk" perspective with negligible computational overhead. Based on this, we investigate DefensiveKV and its layer-aware variant, Layer-DefensiveKV, both of which achieve significant improvements over state-of-the-art methods across comprehensive evaluations. Our work pioneers a new research direction by emphasizing the "worst-case risk"-aware aggregation to mitigate the often-overlooked fragility in cache eviction—a critical yet underexplored component of efficient LLM inference. We hope these contributions pave the way for more effective cache eviction methods, which are essential for advancing LLM inference.

## ACKNOWLEDGE

We would like to thank the anonymous reviewers for their valuable feedback and constructive comments. This work was supported by the National Natural Science Foundation of China (NSFC) under Grants 62472400 and 62271465, the National Key R&D Program of China under Grant 2025YFC3408300, and the Suzhou Basic Research Program under Grant SYG202338.

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

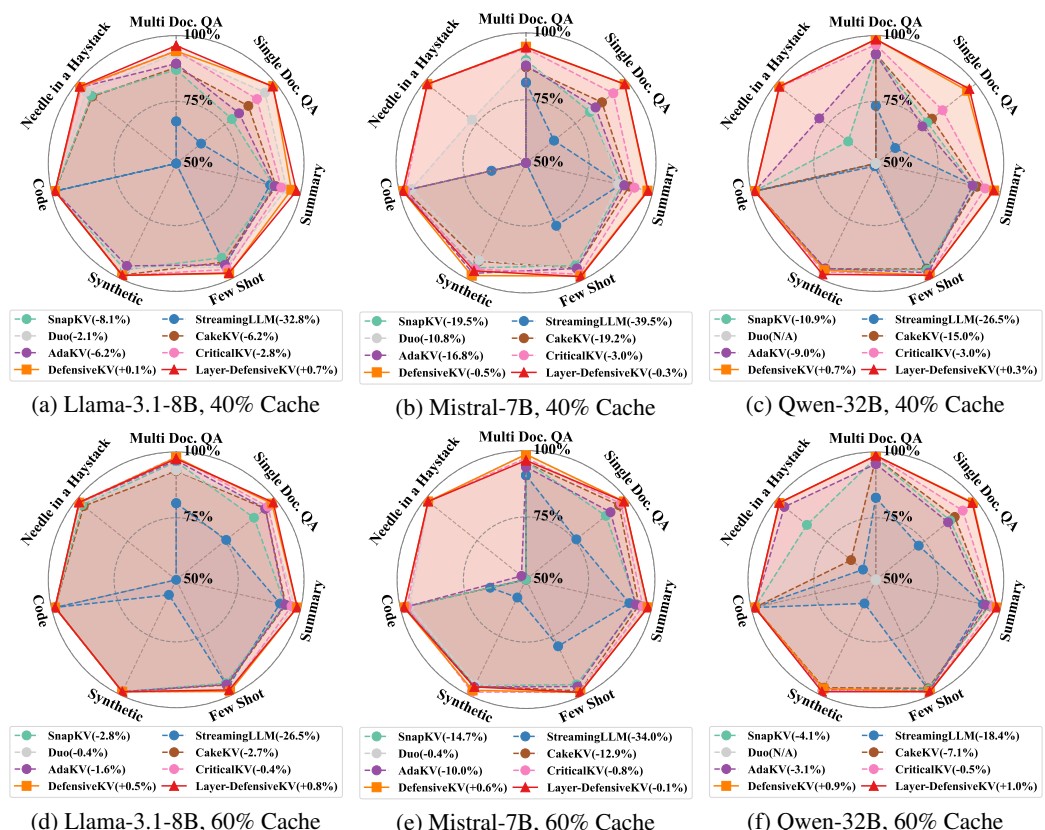

Figure 10: Summarization of quality losses.

# A    QUALITY LOSSES OF METHODS WITH 40% AND 60% CACHE SIZE

Figure 10 further summarizes the quality losses of different methods at 40% and 60% cache sizes. It can be observed that both DefensiveKV and Layer-DefensiveKV maintain nearly lossless performance, in some cases even surpassing the original uncompressed results. In contrast, all other methods exhibit notable declines in quality. These results demonstrate the effectiveness of our approach.

# B    DETAILED SETTINGS

The fundamental settings for SnapKV, CAKE, AdaKV, CriticalKV and our methods were kept as originally defined, with an average-pooling kernel size of 5 and a historical token size of 32 for observation. For StreamingLLM (Xiao et al., 2023), we follow standard settings, using 4 sink tokens and retaining the most recent window's cache. For DuoAttention (Xiao et al., 2025), we follow the publicly released training settings. Following standard practices in prior studies (Li et al., 2024; Zhang et al., 2024a; Feng et al., 2024), we perform cache eviction immediately after the prefilling phase of each layer.

# C    THEORETICAL ANALYSIS OF DEFENSIVE AGGREGATION

To formalize our analysis of defensive aggregation versus mean aggregation, we first establish a mathematical model that captures the risk of fragile stability. We then derive the statistical properties of the risk estimators resulting from each aggregation method. Therefore, to formally analyze this specific failure case, we construct the following assumption:

## C.1   A MATHEMATICAL MODEL FOR WORST-CASE RISK UNDER FRAGILE STABILITY

**Definition 1 (Risk Target)** *As introduced in Section 3.3, we define the True Risk $R_i$ of evicting cache entry $i$ as its peak importance over all future generation steps $L$. Thus, the objective of a risk estimator is to estimate this quantity:*

$$R_i = \sup_{t \in L} I_{t,i}$$

An ideal eviction policy based on worst-case risk control should retain the top cache entries with the highest True Risk $R_i$. Our analysis focuses on modeling the performance of different aggregation strategies on worst-case risk control. Therefore, we must model the failure cases in estimating worst-case risk that arise from the "Fragile Stability" phenomenon. This fragility can manifest in two primary ways: (1) an entry that is "stably high" in importance suddenly drops, or (2) an entry that is "stably low" in importance suddenly spikes.

The first case (stably high) is not the critical failure mode for our analysis. Both Mean Aggregation and Defensive Aggregation will correctly identify such entries as high-risk on average and retain them. The critical failure is the second case: an entry appears stable but unimportant for long durations, yet experiences rare, sudden spikes of high importance. In this case, Mean Aggregation fails in this scenario because its expectation is anchored to the common low-importance state, causing it to severely underestimate the true risk $R_i$. Our model is constructed specifically to analyze this case.

**Definition 2 (Critical Failure Case Model)** *We model the "fragile" importance distribution $F_i$ for entry $i$ as a Bernoulli mixture model defined by two component distributions:*

- *Stable (Low-Importance) Distribution ($F_i^{low}$): Activated with probability $1 - \epsilon_i$. Observations are $I_{j,i} \sim F_i^{low}$, with $E[I|F_i^{low}] = \mu_i^{low}$ and $\sup(F_i^{low}) = R_i^{low}$.*

- *Fragile (High-Importance) Distribution ($F_i^{high}$): Activated with probability $\epsilon_i$. Observations are $I_{j,i} \sim F_i^{high}$, with $E[I|F_i^{high}] = \mu_i^{high}$ and $\sup(F_i^{high}) = R_i$.*

*The "fragility" is mathematically defined by two properties:*

1. *Sparsity: The risk event is rare, $\epsilon_i \ll 0.5$.*

2. *Extremity: The risk event is severe, $R_i \geq \mu_i^{high} \gg \mu_i^{low}$.*

## C.2   THEOREMS AND PROOFS

**Theorem 1 (Estimation Bias of Mean Aggregation)** *Under Definition 1 and Definition 2, the Mean Aggregation estimator, $S_i^{mean} = \frac{1}{m}\sum_{j=1}^{m} I_{j,i}$, is a high-bias estimator of the true risk $R_i$. This bias is systematic and worsens as the risk event's sparsity increases ($\epsilon_i \to 0$).*

**Proof 1** *By the definition of the mixture model (Definition 2), the expected value of a single observation $I_{j,i}$ is:*

$$E[I_{j,i}] = (1 - \epsilon_i)E[I|F_i^{low}] + \epsilon_i E[I|F_i^{high}] = (1 - \epsilon_i)\mu_i^{low} + \epsilon_i \mu_i^{high}$$

*By the linearity of expectation, the expected value of the Mean Aggregation estimator $S_i^{mean}$ is:*

$$E[S_i^{mean}] = E\left[\frac{1}{m}\sum_{j=1}^{m} I_{j,i}\right] = \frac{1}{m}\sum_{j=1}^{m} E[I_{j,i}] = (1 - \epsilon_i)\mu_i^{low} + \epsilon_i \mu_i^{high}$$

*The bias of this estimator with respect to the true risk $R_i$ (from Definition 1) is:*

$$Bias(S_i^{mean}) = R_i - E[S_i^{mean}] = R_i - \left[(1 - \epsilon_i)\mu_i^{low} + \epsilon_i \mu_i^{high}\right]$$

*Given Definition 2, $R_i \geq \mu_i^{high}$ and $\epsilon_i$ is small, meaning the value of $E[S_i^{mean}]$ is dominated by $\mu_i^{low}$. In the limit as $\epsilon_i \to 0$ (the "most fragile" case), $E[S_i^{mean}] \to \mu_i^{low}$. The bias thus approaches:*

$$\lim_{\epsilon_i \to 0} Bias(S_i^{mean}) = R_i - \mu_i^{low}$$

*This is a large negative value, indicating a severe underestimation of the true risk. The mean aggregation estimator is mathematically anchored to the mean of the low-importance mode, $\mu_i^{low}$, not the peak risk $R_i$.*

**Theorem 2 (Lower Bias of Worst-case Risk Estimation Defensive Aggregation)** *Under Definition 1 and Definition 2, the core component of Defensive Aggregation (Algorithm 1), the worst-case risk estimator $\tilde{R}_i = \max_{1 \leq j \leq m} I_{j,i}$, is a significantly lower-bias estimator of $R_i$ than Mean Aggregation.*

**Proof 2** *The estimator $\tilde{R}_i$ aims to capture the high-importance mode, $F_i^{high}$. We first analyze its probability of success. $\tilde{R}_i$ fails to capture the risk signal if and only if all $m$ independent observations are drawn from the low-importance mode $F_i^{low}$.*

$$P(\text{Failure}) = P(\tilde{R}_i \leq R_i^{low}) = P(I_{j,i} \sim F_i^{low} \text{ for all } j = 1, \ldots, m) = (1 - \epsilon_i)^m$$

*The probability of successfully capturing at least one high-importance signal is:*

$$P(\text{Success}) = P(\tilde{R}_i > R_i^{low}) = 1 - (1 - \epsilon_i)^m$$

*We now compare the bias by examining the exact expectations. From Theorem 1:*

$$E[S_i^{mean}] = (1 - \epsilon_i)\mu_i^{low} + \epsilon_i\mu_i^{high}$$

*For the Defensive Aggregation estimator $\tilde{R}_i$, we establish a lower bound. The expectation $E[\tilde{R}_i]$ is:*

$$E[\tilde{R}_i] = P(\text{Failure}) \cdot E[\max(I)|\text{Fail}] + P(\text{Success}) \cdot E[\max(I)|\text{Success}]$$

*We know $E[\max(I)|F_i^{low}] \geq E[I|F_i^{low}] = \mu_i^{low}$ and $E[\max(I)|\text{Success}] \geq E[I|F_i^{high}] = \mu_i^{high}$, then substituting these bounds:*

$$E[\tilde{R}_i] \geq (1 - \epsilon_i)^m \mu_i^{low} + (1 - (1 - \epsilon_i)^m)\mu_i^{high}$$

*To compare this to $E[S_i^{mean}]$, let us define a function $g(p) = (1 - p)\mu_i^{low} + p\mu_i^{high}$. By Definition 2, $\mu_i^{high} > \mu_i^{low}$, so $g(p)$ is a strictly increasing function of $p$. We can now write the expectations in terms of $g(p)$:*

- $E[S_i^{mean}] = g(\epsilon_i)$

- $E[\tilde{R}_i] \geq g(1 - (1 - \epsilon_i)^m)$

*Since $g$ is strictly increasing, this is equivalent to showing $1 - (1 - \epsilon_i)^m > \epsilon_i$. Considering $m \geq 2$ and $\epsilon_i \in (0, 1)$, this clearly holds, which strictly proves that $E[\tilde{R}_i] > E[S_i^{mean}]$.*

*Both estimators are negatively biased ($E[\text{estimator}] \leq R_i$), thus*

$$0 \leq Bias(\tilde{R}_i) = R_i - E[\tilde{R}_i] < R_i - E[S_i^{mean}] = Bias(S_i^{mean})$$

*Therefore, the worst-case estimator $\tilde{R}_i$ in defensive aggregation has a smaller bias.*

**Corollary 1 (Robustness from Adaptive Prior-Risk Correction)** *The Adaptive Prior-Risk Correction step in Algorithm 1, $R_i = \max(\tilde{R}_i, \bar{R})$, provides robustness against the primary failure mode of $\tilde{R}_i$: under-observation.*

According to the Proof of Theorem 2, we can derive that:

$$Bias(\tilde{R}_i) = R_i - E[\tilde{R}_i] = R_i - P(\text{Failure}) \cdot E[\max(I)|\text{Fail}] - P(\text{Success}) \cdot E[\max(I)|\text{Success}]$$

$$Bias(\tilde{R}_i) = R_i - (1 - \epsilon_i)^m \cdot E[\max(I)|\text{Fail}] - (1 - (1 - \epsilon_i)^m) \cdot E[\max(I)|\text{Success}]$$

This highlights a weakness of $\tilde{R}_i$: if the observation $m$ is too small relative to the sparsity $\epsilon_i$ (i.e., $m\epsilon_i \ll 1$), the failure probability $(1 - \epsilon_i)^m$ remains high. This corresponds to the "limited observation" problem (Section 3.3). In this failure case, all observations are from $F_i^{low}$, and the bias of estimator becomes $Bias(\tilde{R}_i) \approx R_i - E[\max(I)|\text{Fail}] \approx R_i - R_i^{low}$, which is a severe underestimation.

Our Adaptive Prior-Risk Correction step, $R_i = \max(\tilde{R}_i, \bar{R})$, replaces this underestimated value $\tilde{R}_i \approx R_i^{low}$ with the adaptive prior $\bar{R}$ (the head-level average risk). This is mathematically equivalent to a Bayesian posterior estimation: when our specific observation for entry $i$ is insufficient, we distrust the observation $\tilde{R}_i$ and revert to our prior knowledge $\bar{R}$.

Table 4: Detailed scores of 16 datasets on LongBench (Llama-3.1-8B-Instruct, 10% Cache Size). The best result is highlighted in **bold**.

| Method | Single-Document QA | | | Multi-Document QA | | | Summarization | | | Few-shot Learning | | | Synthetic | | Code | | Avg. |
|---|---|---|---|---|---|---|---|---|---|---|---|---|---|---|---|---|---|
| | NrtvQA | Qasper | MF-en | Hotpot | 2WikiQA | Musique | GovRep | QMSum | MultiNews | TREC | TriviaQA | SAMSum | PCount | PR-en | Lcc | RB-P | |
| Llama-3.1-8B-Instruct, 10% Cache Size | | | | | | | | | | | | | | | | | |
| Full Cache | 29.55 | 44.68 | 55.82 | 57.59 | 48.89 | 32.61 | 34.40 | 25.51 | 26.83 | 73.00 | 92.36 | 43.27 | 7.38 | 99.50 | 63.44 | 52.36 | 49.20 |
| SnapKV | 23.25 | 20.77 | 23.54 | 43.11 | 22.29 | 18.57 | 24.13 | 19.51 | 20.43 | 43.00 | 92.01 | 43.18 | 7.35 | 51.50 | 64.80 | 54.17 | 35.73 (↓27.4%) |
| CAKE | 24.25 | 22.79 | 24.69 | 44.49 | 23.58 | 20.30 | 24.70 | 20.75 | 20.38 | 44.50 | **92.36** | 43.44 | 5.15 | 76.00 | 65.52 | 52.95 | 37.87 (↓23.0%) |
| AdaKV | 24.54 | 22.69 | 25.47 | 41.86 | 22.59 | 16.74 | 24.52 | 20.17 | 20.81 | 46.00 | 92.25 | 43.36 | 7.23 | 50.50 | 66.25 | 54.84 | 36.24 (↓26.3%) |
| CriticalKV | **27.62** | 25.02 | 28.78 | 43.49 | 23.50 | 20.67 | 25.69 | 21.39 | 21.10 | 48.50 | 92.17 | 42.84 | 7.30 | 63.50 | 66.61 | 54.09 | 38.27 (↓22.2%) |
| DefensiveKV | 27.23 | 25.32 | 35.30 | 46.29 | 27.15 | 18.23 | 25.42 | 22.39 | 20.88 | **54.00** | 91.96 | **44.13** | 5.90 | **98.50** | 67.46 | 54.15 | 41.52 (↓15.6%) |
| Layer-DefensiveKV | 27.30 | **28.53** | **38.09** | **49.12** | **30.80** | **22.44** | **25.76** | **22.42** | **21.41** | 52.00 | 92.01 | 43.94 | **8.68** | 98.00 | **67.66** | **56.01** | **42.76** (↓13.1%) |

Table 5: Detailed scores of LongBench on MoE LLMs (Qwen-3-30B-A3B, 20% Cache Size). The best result among compressed methods is highlighted in **bold**.

| Method | Single-Document QA | | | Multi-Document QA | | | Summarization | | | Few-shot Learning | | | Synthetic | | Code | | Avg. |
|---|---|---|---|---|---|---|---|---|---|---|---|---|---|---|---|---|---|
| | NrtvQA | Qasper | MF-en | Hotpot | 2WikiQA | Musique | GovRep | QMSum | MultiNews | TREC | TriviaQA | SAMSum | PCount | PR-en | Lcc | RB-P | |
| Qwen-3-30B-A3B, 20% Cache Size | | | | | | | | | | | | | | | | | |
| Full Cache | 23.62 | 42.12 | 54.49 | 63.83 | 56.70 | 32.74 | 30.90 | 22.04 | 23.59 | 77.00 | 91.11 | 45.82 | 15.50 | 100.00 | 72.27 | 69.77 | 51.34 |
| CriticalKV (Stongest Baseline) | 23.80 | 33.37 | 38.51 | 54.73 | 41.18 | 26.92 | **29.70** | 20.49 | 20.64 | 68.00 | **90.38** | 45.07 | 15.00 | 99.50 | 73.11 | 70.59 | 46.94 (↓8.6%) |
| DefensiveKV | **24.19** | 36.66 | 42.75 | **59.14** | 44.91 | **31.95** | 29.56 | 21.13 | 20.94 | 69.50 | 90.18 | **45.58** | 15.00 | **100.00** | **73.16** | 70.55 | 48.45 (↓5.6%) |
| Layer-DefensiveKV | 22.98 | **38.99** | **46.30** | 58.50 | **50.02** | 31.69 | 29.17 | **21.54** | **21.33** | **74.00** | 90.29 | 45.32 | **16.00** | 100.00 | 72.73 | **71.10** | **49.37** (↓3.8%) |

## D  EVALUATION UNDER MORE EXTREME COMPRESSION

In our main experiments, we initially defined a 20% cache budget as an extreme compression scenario in real-world benchmark. Under this condition, our proposed DefensiveKV and Layer-DefensiveKV demonstrated minimal performance degradation on the Llama-3.1-8B model, with average quality losses of just 4.8% and 2.6%, respectively. In contrast, the strongest baseline, CriticalKV, suffers an 11.1% performance drop, while other baselines degrade even more significantly.

To push the limits further, we evaluated our method with the cache budget reduced to 10% on LongBench. As presented in Table 4, the results underscore the robustness of our approach. Layer-DefensiveKV maintained a high average score of 42.76, a modest 13.1% drop from the full-cache performance. This result significantly outperforms all baselines, including the most competitive one, CriticalKV, which scored 38.27 (a 22.2% drop). This provides further evidence that our defensive aggregation strategy remains highly effective even under severely constrained cache conditions.

## E  GENERALIZATION ANALYSIS ON MIXTURE-OF-EXPERTS (MOE) ARCHITECTURES

To demonstrate the broad applicability of our approach, we extend our evaluation to the Mixture-of-Experts (MoE) architecture, which has become increasingly prevalent in recent state-of-the-art LLMs. While MoE architectures introduce sparsity within the Feed-Forward Networks (FFN) to scale parameters efficiently, the attention mechanism—where the KV cache resides—typically retains the same dense structure as standard Transformer models. Since the cache eviction methods operates within the attention blocks, it is agnostic to the modification of MoE in FFN structure. Therefore, our defensive aggregation strategy inherently generalize to MoE models without requiring any architecture-specific modifications.

We conducted experiments on the Qwen-3-30B-A3B model Team (2025), a powerful open-source MoE LLMs. We evaluated the performance on the LongBench under a strict 20% KV cache budget. As shown in Table 5, the results confirm the effectiveness of our method in the MoE setting: our methods consistently outperform the strong baseline, CriticalKV. Specifically, **Layer-DefensiveKV** limits the average performance degradation to only **3.8%** compared to the Full Cache upper bound, whereas CriticalKV suffers a significantly larger drop of **8.6%**.

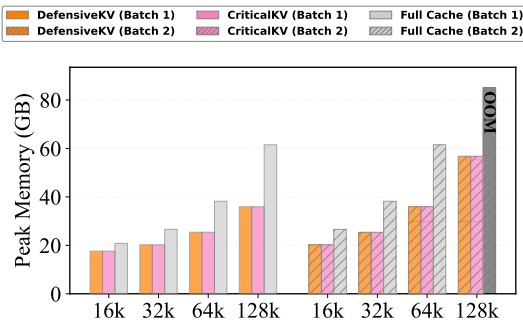

Figure 11: Peak Memory usage(All with FlashAttention-2).

## F    ADDITIONAL RELATED WORKS

Beyond cache eviction methods, a broader range of related work can reduce KV cache footprint. For example, Think (Xu et al., 2024) compresses the KV cache by reducing the number of channels in the key states. Other approaches, such as MiniCache (Liu et al., 2024a) and KVSharer (Yang et al., 2024), exploit KV similarity between layers to achieve compression. These techniques are orthogonal to KV cache eviction methods and can be further combined with them. KV cache quantization (Hooper et al., 2024; Liu et al., 2024d), which reduces the precision of individual cache entries (e.g., quantizing 16-bit entries to 4-bit or 2-bit), also offers footprint reduction. Because quantization methods typically retain all cache entries, they are fundamentally orthogonal to the cache eviction methods explored in this paper and can also be applied to further enhance them. Furthermore, recent speculative decoding methods explore using a reduced KV cache for draft generation in long-sequence generation (Sun et al., 2024; Yang et al., 2025). Refining cache eviction to enhance speculative decoding is also a promising research direction.

Sparse attention methods are conceptually related to KV cache eviction (Xiao et al., 2024a; Tang et al., 2024; Jiang et al., 2024; Liu et al., 2024b). The key difference is that KV cache eviction retains only a subset of the KV cache, while sparse attention methods keep all entries but selectively utilize only a critical subset during computation (Nawrot et al., 2025). Consequently, sparse attention methods do not reduce the memory footprint of the KV cache. The two technique lines are, in fact, orthogonal. Future research could explore firstly employing KV cache eviction to compress the cache to a certain proportion (e.g., 40% cache size with minimal loss, as demonstrated in this paper) and then applying sparse attention for further acceleration. This represents a promising direction for future research.

## G    MEMORY USAGE DURING GENERATION

Following the efficiency evaluation in Section 4.5, we also measured peak memory usage during inference. The memory savings from cache eviction are primarily determined by the compressed cache size. Our introduced defensive aggregation method does not differ in memory usage from standard mean aggregation. As shown in Figure 11, DefensiveKV and CriticalKV exhibit significantly lower peak memory usage than Full Cache. For example, with a batch size of 1 and a 128K context length, DefensiveKV and CriticalKV use only 36GB, far less than Full Cache's 61.5 GB. This allows them to support larger batch sizes, such as batch size 2, further increasing decoding throughput, while Full Cache encounters out-of-memory (OOM) errors. This advantage enables DefensiveKV and CriticalKV to achieve up to a 4.2x speedup in 128K decoding throughput compared to Full Cache.

## H    INTEGRATION OF DEFENSIVEKV WITH ORTHOGONAL EFFICIENT INFERENCE TECHNIQUES

Both existing KV cache eviction methods and our enhanced method, DefensiveKV, can be seamlessly integrated with orthogonal efficient inference techniques to achieve even greater memory savings and computational efficiency. Such techniques include KV cache quantization (applying eviction then

Table 6: Detailed scores of 13 datasets on Ruler.

| Method | cwe | fwe | niah_mk1 | niah_mk2 | niah_mk3 | niah_mq | niah_mv | niah_s1 | niah_s2 | niah_s3 | qa_1 | qa_2 | vt | Avg. |
|---|---|---|---|---|---|---|---|---|---|---|---|---|---|---|
| Llama-3.1-8B-Instruct, 32K Ruler, 20% Cache Size | | | | | | | | | | | | | | |
| FullCache | 45.22 | 94.13 | 99.60 | 99.60 | 99.40 | 98.75 | 99.10 | 100.00 | 100.00 | 100.00 | 79.80 | 54.80 | 99.24 | 89.97 |
| Strongest Baseline CriticalKV | **26.80** | 88.80 | 91.60 | 29.40 | 19.40 | 95.00 | 93.60 | 100.00 | 99.60 | 42.40 | 40.80 | 40.20 | 97.76 | 66.57 |
| DefensiveKV | 22.94 | 90.00 | **99.80** | 86.80 | 97.00 | 98.65 | 97.90 | 100.00 | 100.00 | 97.40 | 68.80 | 45.80 | **98.76** | 84.91 |
| Layer-DefensiveKV | 17.86 | **90.80** | 99.60 | **99.40** | **99.00** | **98.85** | **98.45** | 100.00 | 100.00 | 100.00 | **73.00** | **47.60** | 98.56 | **86.39** |
| Llama-3.1-8B-Instruct, 32K Ruler, 40% Cache Size | | | | | | | | | | | | | | |
| FullCache | 45.22 | 94.13 | 99.60 | 99.60 | 99.40 | 98.75 | 99.10 | 100.00 | 100.00 | 100.00 | 79.80 | 54.80 | 99.24 | 89.97 |
| Strongest Baseline CriticalKV | 49.08 | 91.93 | **99.60** | 94.00 | 54.00 | **98.75** | 98.90 | 100.00 | 100.00 | 97.40 | 68.00 | 47.60 | **99.32** | 84.51 |
| DefensiveKV | **51.12** | **92.87** | **99.60** | **99.80** | 98.60 | 98.65 | 98.90 | 100.00 | 100.00 | 100.00 | **78.20** | 51.80 | 99.24 | 89.91 |
| Layer-DefensiveKV | 50.24 | 92.00 | **99.60** | **99.80** | **99.20** | 98.70 | **99.05** | 100.00 | 100.00 | 100.00 | 78.00 | **53.40** | 99.12 | **89.93** |

quantization to leverage both cache sparsity and precision reduction), KV cache retrieval (compressing the cache before retrieval to reduce overhead), and KV cache offloading (evicting before offloading to reduce storage and transfer costs in slower memory systems).

As a case study, we demonstrate DefensiveKV integration with KV cache quantization. We adopted the official HuggingFace-provided int4 quantization (Face, 2024a) with Quanto backend support (Face, 2024b). We first use DefensiveKV to retain 40% of cache entries, then apply 4-bit quantization, yielding 10% of the original memory usage. Moreover, we also present results for DefensiveKV retaining only 10% of cache entries without quantization.

As shown in Table 8, combining DefensiveKV with int4 quantization to reduce the cache to 10% of its original memory footprint achieves an average score of 48.55, incurring less than a one-point drop compared to the full-cache baseline (49.20). This is not achievable with quantization alone, which would require approximately 1.6-bit quantization. It also substantially outperforms using cache eviction alone to reduce the cache size to 10%. Thus, combining DefensiveKV and quantization substantially outperforms either technique individually, highlighting strong potential in real-world deployment.

# I MORE NEEDLE-IN-A-HAYSTACK-STYLE EVALUATIONS ON RULER BENCHMARK

In the Needle-in-A-Haystack task, a keyword, referred to as the "needle", is embedded within a lengthy context known as the "haystack". The objective of this task is to extract the "needle" from the "haystack", which is composed of essays by Paul Graham (Kamradt, 2023). In our main experiments, we adopt the respective prompt templates (see Table 13) used in the Ruler Benchmark (Hsieh et al., 2024) (corresponding to NIAH-s2 and NIAH-MV in their formulation) to ensure consistency and reproducibility

The whole Ruler benchmark (Hsieh et al., 2024) comprises 13 synthetic, needle-in-a-haystack-style tasks designed to evaluate the long-context capabilities of models. A single evaluation on the full 32K RULER benchmark requires approximately 9 GPU hours. Consequently, a comprehensive assessment across all methods, compression rates, and models would demand an estimated 864 GPU hours, which is computationally prohibitive.

In this section, we further presents a more extensive analysis on the complete Ruler benchmark. We evaluated our proposed methods and the strongest baseline, CriticalKV, at 20% and 40% cache sizes using Llama-3.1-8B-Instruct, with the results detailed in Table 6. Both DefensiveKV and Layer-DefensiveKV demonstrated significant advantages; for instance, at a 20% cache size, they achieved average scores of 84.91 and 86.39, respectively, substantially outperforming the CriticalKV baseline's score of 66.57. These findings underscore our method's ability to achieve strong compression performance with minimal loss in accuracy.

Table 7: Performance comparison with and without defensive aggregation on LongBench.

| Method | Single-Document QA | | | Multi-Document QA | | | Summarization | | | Few-shot Learning | | | Synthetic | | Code | | Avg. |
|---|---|---|---|---|---|---|---|---|---|---|---|---|---|---|---|---|---|
| | NrtvQA | Qasper | MF-en | Hotpot | 2WikiQA | Musique | GovRep | QMSum | MultiNews | TREC | TriviaQA | SAMSum | PCount | PR-en | Lcc | RB-P | |
| Llama-3.1-8B-Instruct, 20% Cache Size | | | | | | | | | | | | | | | | | |
| Full Cache | 29.55 | 44.68 | 55.82 | 57.59 | 48.89 | 32.61 | 34.40 | 25.51 | 26.83 | 73.00 | 92.36 | 43.27 | 7.38 | 99.50 | 63.44 | 52.36 | 49.20 |
| AdaKV | 27.07 | 28.69 | 32.85 | 49.64 | 30.89 | 21.57 | 26.70 | 21.85 | 22.67 | 55.50 | 91.30 | 43.89 | 7.30 | 80.50 | 66.44 | 55.43 | 41.39 |
| + Defensive | **28.60** | **37.62** | **41.08** | **51.74** | **36.87** | **22.83** | **27.83** | **23.18** | **23.51** | **66.00** | **91.64** | **44.35** | **8.10** | **92.50** | **67.97** | **55.71** | **44.97** |
| CriticalKV | 29.81 | 32.58 | 34.96 | **52.34** | 36.24 | 26.37 | 28.35 | 23.52 | 23.24 | 56.50 | 90.80 | **43.37** | **8.89** | 93.00 | 67.05 | 54.99 | 43.88 |
| + Defensive | **29.97** | **40.46** | **46.23** | 52.20 | **38.40** | **28.06** | **29.96** | **23.89** | **24.11** | **68.00** | **91.58** | 43.17 | 8.28 | **100.00** | **67.17** | **55.40** | **46.68** |

Table 8: Performance of DefensiveKV combined with int4 cache quantization on LongBench

| Method | Single-Document QA | | | Multi-Document QA | | | Summarization | | | Few-shot Learning | | | Synthetic | | Code | | Avg. |
|---|---|---|---|---|---|---|---|---|---|---|---|---|---|---|---|---|---|
| | NrtvQA | Qasper | MF-en | Hotpot | 2WikiQA | Musique | GovRep | QMSum | MultiNews | TREC | TriviaQA | SAMSum | PCount | PR-en | Lcc | RB-P | |
| Llama-3.1-8B-Instruct, 20% Cache Size | | | | | | | | | | | | | | | | | |
| Full Cache (100% memory) | 29.55 | 44.68 | 55.82 | 57.59 | 48.89 | 32.61 | 34.4 | 25.51 | 26.83 | 73 | 92.36 | 43.27 | 7.38 | 99.5 | 63.44 | 52.36 | 49.2 |
| DefensiveKV-40% Cache (40% memory) | 30.07 | 46.37 | 54.9 | 57.5 | 45.97 | 28.85 | 33.7 | 24.69 | 26.2 | 71.5 | 91.78 | 43.69 | 9.88 | 100 | 66.25 | 55.97 | 49.21 |
| DefensiveKV-10% Cache (10% memory) | 27.23 | 25.32 | 35.30 | 46.29 | 27.15 | 18.23 | 25.42 | 22.39 | 20.88 | 54.00 | **91.96** | **44.13** | 5.90 | 98.50 | **67.46** | 54.15 | 41.52 |
| DefensiveKV-40% Cache-int4 (10% memory) | **30.63** | **44.62** | **54.44** | **56.14** | **42.9** | **28.15** | **33.79** | **25.15** | **25.92** | **70.5** | 91.28 | 43.76 | **7.55** | **100** | 65.71 | **56.23** | **48.55** |

## J CASE STUDY: AUGMENTING EVICTION METHODS VIA DEFENSIVE AGGREGATION

Our proposed defensive aggregation is an orthogonal strategy that is compatible with other cache eviction methods and can be seamlessly integrated in a plug-and-play manner. In the main paper, we demonstrate its application to CriticalKV. Furthermore, we extend the defensive aggregation to AdaKV. As shown in the table 7, defensive aggregation effectively improved both AdaKV's and CriticalKV's performance, increasing the average score from 41.39 to 44.97 and from 43.88 to 46.68, respectively. These results indicate that defensive aggregation can broadly enhance existing cache eviction methods.

## K DETAILED SCORES OF LONGBENCH

We provide detailed scores on individual datasets for 40%, 60% and 80% cache sizes in Tables 9, 10 and, 11. Our DefensiveKV and Layer-DefensiveKV methods maintain nearly lossless generation quality across these settings, while other baselines fail to achieve this level of performance.

## L THE EFFECTIVENESS OF DEFENSIVE AGGREGATION STRATEGY

To complement Figure 1 in the main text, Figure 12 provides additional visualizations demonstrating that defensive aggregation offers greater robustness than mean aggregation under a 50% cache size. The results reveal that this fragility is prevalent across numerous layers. Both "Single Historical token" and "Mean aggregation" methods exhibit high sensitivity to this fragility, leading to poor worst-case performance. In contrast, defensive aggregation effectively mitigates this issue, consistently maintaining higher worst-case values.

## M FURTHER ELABORATION OF THE FRAGILE STABILITY ASSUMPTION

Complementing Figure 3 in the main text, Figure 13 provides a more detailed illustration. It demonstrates how measurements from single historical tokens, which guide cache eviction, experience significant degradation at certain generation steps. The outlier cases occurs regardless of which specific historical token is used. Consequently, the failure of such averaging approaches is an expected outcome.

Table 9: Detailed scores of 16 datasets on LongBench (40% cache size).

| Method | Single-Document QA | | | Multi-Document QA | | | Summarization | | | Few-shot Learning | | | Synthetic | | Code | | Avg. |
| | NrtvQA | Qasper | MF-en | Hotpot | 2WikiQA | Musique | GovRep | QMSum | MultiNews | TREC | TriviaQA | SAMSum | PCount | PR-en | Lcc | RB-P | |
|---|---|---|---|---|---|---|---|---|---|---|---|---|---|---|---|---|---|
| *Llama-3.1-8B-Instruct, 40% Cache Size* | | | | | | | | | | | | | | | | | |
| Full Cache | 29.55 | 44.68 | 55.82 | 57.59 | 48.89 | 32.61 | 34.40 | 25.51 | 26.83 | 73.00 | 92.36 | 43.27 | 7.38 | 99.50 | 63.44 | 52.36 | 49.20 |
| DuoAttention | 28.83 | 42.51 | 53.35 | 55.76 | 45.37 | 30.16 | 32.26 | 25.07 | 25.51 | 71.50 | 88.44 | 41.12 | 3.67 | 99.50 | 68.81 | 58.85 | 48.17 |
| StreamingLLM | 24.32 | 28.77 | 28.74 | 43.75 | 30.71 | 18.55 | 30.26 | 21.86 | 24.88 | 65.50 | 92.24 | 41.65 | 2.92 | 46.00 | 66.54 | 61.42 | 39.26 |
| SnapKV | 27.72 | 36.42 | 38.31 | 54.92 | 40.02 | 26.95 | 30.60 | 23.33 | 24.31 | 56.00 | 92.31 | 43.92 | 7.62 | 96.50 | 65.95 | 53.77 | 44.92 |
| CAKE | 30.43 | 37.57 | 45.51 | 57.13 | 40.08 | 25.95 | 30.33 | 23.80 | 24.96 | 61.00 | 91.83 | 43.46 | 6.70 | 100.00 | 65.56 | 52.66 | 46.06 |
| AdaKV | 28.36 | 37.58 | 41.35 | 54.80 | 41.47 | 29.02 | 30.18 | 23.72 | 24.68 | 63.50 | 91.73 | 43.57 | 7.27 | 95.00 | 64.93 | 54.75 | 45.74 |
| CriticalKV | 30.10 | 40.14 | 49.03 | 55.95 | 46.22 | 30.42 | 31.49 | 24.34 | 25.15 | 67.50 | 92.39 | 43.20 | 8.08 | 99.00 | 64.68 | 55.08 | 47.67 |
| DefensiveKV | 30.07 | 46.37 | 54.90 | 57.50 | 45.97 | 28.85 | 33.70 | 24.69 | 26.20 | 71.50 | 91.78 | 43.69 | 9.88 | 100.00 | 66.25 | 55.97 | 49.21 |
| Layer-DefensiveKV | 30.94 | 43.84 | 55.01 | 56.36 | 49.14 | 29.88 | 34.09 | 25.71 | 26.64 | 72.00 | 91.49 | 42.96 | 8.56 | 99.50 | 67.30 | 57.98 | 49.46 |
| *Mistral-7B-Instruct-v0.3, 40% Cache Size* | | | | | | | | | | | | | | | | | |
| Full Cache | 27.02 | 38.19 | 50.22 | 50.75 | 37.41 | 27.92 | 34.45 | 25.76 | 26.37 | 76.00 | 89.01 | 46.89 | 6.50 | 97.00 | 66.04 | 60.47 | 47.50 |
| DuoAttention | 20.37 | 26.96 | 49.69 | 48.92 | 34.96 | 20.16 | 29.14 | 21.74 | 24.86 | 73.50 | 87.39 | 44.06 | 3.00 | 93.00 | 63.95 | 58.16 | 43.74 |
| StreamingLLM | 19.71 | 24.85 | 29.54 | 42.10 | 34.34 | 18.53 | 31.03 | 21.60 | 24.05 | 40.00 | 83.25 | 41.22 | 3.50 | 45.50 | 34.40 | 46.59 | 33.76 |
| SnapKV | 25.32 | 30.09 | 39.68 | 49.16 | 33.66 | 22.38 | 30.60 | 22.39 | 24.22 | 65.00 | 89.37 | 47.17 | 5.00 | 94.50 | 65.60 | 60.66 | 44.05 |
| CAKE | 25.02 | 31.82 | 45.30 | 48.42 | 31.94 | 21.73 | 31.70 | 23.15 | 24.77 | 68.50 | 89.22 | 46.34 | 4.00 | 92.50 | 64.99 | 60.21 | 44.35 |
| AdaKV | 24.76 | 31.86 | 41.79 | 49.59 | 32.95 | 20.20 | 30.51 | 22.94 | 24.42 | 68.00 | 88.96 | 47.29 | 5.50 | 96.50 | 65.54 | 59.99 | 44.42 |
| CriticalKV | 26.97 | 34.32 | 47.50 | 48.00 | 38.07 | 24.64 | 31.51 | 24.79 | 25.14 | 72.50 | 89.37 | 47.86 | 4.53 | 95.50 | 65.59 | 59.68 | 46.00 |
| DefensiveKV | 25.45 | 39.24 | 51.42 | 50.13 | 34.89 | 26.36 | 34.43 | 25.42 | 26.17 | 75.50 | 89.21 | 46.59 | 5.05 | 98.00 | 66.54 | 61.65 | 47.25 |
| Layer-DefensiveKV | 26.29 | 40.49 | 50.92 | 48.85 | 36.34 | 26.02 | 34.33 | 25.28 | 26.62 | 76.00 | 89.36 | 46.71 | 3.86 | 97.00 | 66.65 | 62.12 | 47.30 |
| *Qwen2.5-32B-Instruct, 40% Cache Size* | | | | | | | | | | | | | | | | | |
| Full Cache | 30.88 | 46.13 | 52.87 | 63.59 | 59.75 | 38.78 | 32.59 | 24.35 | 24.95 | 72.00 | 88.26 | 47.05 | 12.50 | 100.00 | 49.64 | 34.24 | 48.60 |
| DuoAttention | N/A | N/A | N/A | N/A | N/A | N/A | N/A | N/A | N/A | N/A | N/A | N/A | N/A | N/A | N/A | N/A | N/A |
| StreamingLLM | 23.95 | 25.49 | 28.48 | 49.89 | 44.66 | 24.15 | 30.32 | 20.60 | 24.12 | 68.00 | 87.08 | 45.90 | 11.62 | 46.00 | 57.38 | 28.36 | 38.50 |
| SnapKV | 27.18 | 33.99 | 38.19 | 61.11 | 53.18 | 38.01 | 29.95 | 20.84 | 24.05 | 66.00 | 89.16 | 47.02 | 12.00 | 97.75 | 52.19 | 34.75 | 45.27 |
| CAKE | 28.13 | 35.18 | 39.22 | 60.17 | 56.22 | 36.25 | 30.33 | 21.03 | 23.88 | 66.00 | 88.74 | 46.68 | 10.50 | 98.75 | 52.20 | 35.14 | 45.53 |
| AdaKV | 27.16 | 31.83 | 37.07 | 60.81 | 53.92 | 37.66 | 29.84 | 20.79 | 23.07 | 68.50 | 88.93 | 47.40 | 12.00 | 97.00 | 53.05 | 34.56 | 45.22 |
| CriticalKV | 31.65 | 35.23 | 42.70 | 59.24 | 59.26 | 40.10 | 31.25 | 22.67 | 24.02 | 71.00 | 88.80 | 46.91 | 11.00 | 99.75 | 51.76 | 35.70 | 46.94 |
| DefensiveKV | 30.97 | 43.88 | 51.15 | 65.16 | 63.66 | 42.20 | 32.82 | 23.50 | 24.67 | 74.00 | 88.88 | 47.50 | 10.00 | 99.88 | 53.50 | 34.24 | 49.13 |
| Layer-DefensiveKV | 31.61 | 44.02 | 51.86 | 64.35 | 61.10 | 40.88 | 31.94 | 24.26 | 24.62 | 74.50 | 88.75 | 46.76 | 12.00 | 100.00 | 50.88 | 34.64 | 48.89 |

Table 10: Detailed scores of 16 datasets on LongBench (60% cache size).

| Method | Single-Document QA | | | Multi-Document QA | | | Summarization | | | Few-shot Learning | | | Synthetic | | Code | | Avg. |
| | NrtvQA | Qasper | MF-en | Hotpot | 2WikiQA | Musique | GovRep | QMSum | MultiNews | TREC | TriviaQA | SAMSum | PCount | PR-en | Lcc | RB-P | |
|---|---|---|---|---|---|---|---|---|---|---|---|---|---|---|---|---|---|
| *Llama-3.1-8B-Instruct, 60% Cache Size* | | | | | | | | | | | | | | | | | |
| Full Cache | 29.55 | 44.68 | 55.82 | 57.59 | 48.89 | 32.61 | 34.40 | 25.51 | 26.83 | 73.00 | 92.36 | 43.27 | 7.38 | 99.50 | 63.44 | 52.36 | 49.20 |
| DuoAttention | 28.77 | 43.00 | 54.41 | 55.90 | 46.18 | 28.61 | 33.86 | 25.10 | 26.80 | 72.00 | 91.16 | 43.37 | 10.50 | 99.50 | 66.23 | 56.03 | 48.84 |
| StreamingLLM | 25.21 | 39.92 | 33.53 | 49.72 | 39.98 | 22.62 | 31.51 | 23.12 | 25.91 | 69.50 | 92.27 | 42.75 | 3.08 | 57.50 | 66.27 | 61.75 | 42.79 |
| SnapKV | 28.92 | 40.35 | 48.00 | 56.79 | 48.60 | 30.12 | 32.54 | 24.20 | 25.76 | 64.50 | 91.64 | 44.53 | 8.85 | 99.00 | 65.03 | 53.81 | 47.65 |
| CAKE | 29.99 | 41.87 | 53.09 | 55.39 | 42.83 | 32.17 | 32.12 | 24.87 | 25.61 | 66.50 | 92.50 | 43.37 | 7.96 | 99.50 | 64.11 | 52.62 | 47.78 |
| AdaKV | 30.10 | 43.61 | 51.20 | 56.37 | 49.70 | 30.18 | 32.37 | 24.38 | 25.54 | 66.50 | 91.48 | 43.87 | 8.02 | 99.50 | 63.91 | 54.73 | 48.22 |
| CriticalKV | 30.31 | 43.54 | 52.82 | 57.30 | 49.09 | 31.78 | 33.48 | 25.18 | 26.00 | 72.50 | 91.80 | 43.95 | 7.47 | 99.50 | 64.05 | 54.92 | 48.98 |
| DefensiveKV | 30.88 | 43.20 | 55.17 | 55.85 | 50.17 | 31.84 | 34.79 | 25.29 | 26.84 | 73.00 | 92.14 | 43.28 | 9.05 | 99.50 | 63.88 | 56.02 | 49.43 |
| Layer-DefensiveKV | 29.95 | 44.11 | 56.78 | 57.11 | 47.47 | 32.58 | 34.81 | 25.11 | 26.80 | 72.00 | 91.83 | 43.14 | 11.10 | 99.50 | 64.21 | 55.81 | 49.52 |
| *Mistral-7B-Instruct-v0.3, 60% Cache Size* | | | | | | | | | | | | | | | | | |
| Full Cache | 27.02 | 38.19 | 50.22 | 50.75 | 37.41 | 27.92 | 34.45 | 25.76 | 26.37 | 76.00 | 89.01 | 46.89 | 6.50 | 97.00 | 66.04 | 60.47 | 47.50 |
| DuoAttention | 28.86 | 36.56 | 50.54 | 53.32 | 39.19 | 29.22 | 33.91 | 25.16 | 26.77 | 76.00 | 87.57 | 45.40 | 5.00 | 95.00 | 64.75 | 58.91 | 47.26 |
| StreamingLLM | 21.48 | 30.55 | 35.40 | 46.84 | 36.62 | 22.89 | 32.20 | 22.70 | 25.06 | 44.50 | 82.26 | 41.87 | 3.00 | 57.00 | 34.42 | 47.38 | 36.51 |
| SnapKV | 25.44 | 33.77 | 45.63 | 52.52 | 34.30 | 25.94 | 32.54 | 24.59 | 25.37 | 68.00 | 89.41 | 47.12 | 5.00 | 95.50 | 66.22 | 59.47 | 45.68 |
| CAKE | 27.17 | 36.69 | 48.77 | 50.91 | 38.21 | 23.16 | 33.53 | 23.73 | 26.06 | 75.00 | 88.46 | 46.93 | 5.56 | 95.00 | 66.23 | 60.76 | 46.64 |
| AdaKV | 25.47 | 34.87 | 47.21 | 48.61 | 35.64 | 25.97 | 32.38 | 24.24 | 25.54 | 70.50 | 88.91 | 47.00 | 5.06 | 96.00 | 65.85 | 60.20 | 45.84 |
| CriticalKV | 26.06 | 37.46 | 50.28 | 50.41 | 37.11 | 26.80 | 33.28 | 25.56 | 26.80 | 75.50 | 88.81 | 47.54 | 6.35 | 97.00 | 65.30 | 59.61 | 47.06 |
| DefensiveKV | 27.82 | 39.13 | 51.50 | 50.78 | 38.39 | 27.37 | 34.29 | 25.41 | 26.78 | 76.00 | 89.21 | 46.89 | 4.60 | 98.00 | 66.27 | 61.39 | 47.74 |
| Layer-DefensiveKV | 25.59 | 38.95 | 51.99 | 50.81 | 37.09 | 25.35 | 34.50 | 25.68 | 26.78 | 77.50 | 89.04 | 46.96 | 4.00 | 97.00 | 67.00 | 61.50 | 47.48 |
| *Qwen2.5-32B-Instruct, 60% Cache Size* | | | | | | | | | | | | | | | | | |
| Full Cache | 30.88 | 46.13 | 52.87 | 63.59 | 59.75 | 38.78 | 32.59 | 24.35 | 24.95 | 72.00 | 88.26 | 47.05 | 12.50 | 100.00 | 49.64 | 34.24 | 48.60 |
| DuoAttention | N/A | N/A | N/A | N/A | N/A | N/A | N/A | N/A | N/A | N/A | N/A | N/A | N/A | N/A | N/A | N/A | N/A |
| StreamingLLM | 26.06 | 34.49 | 33.06 | 57.33 | 46.75 | 30.57 | 30.39 | 21.93 | 24.83 | 72.50 | 87.01 | 46.97 | 10.62 | 57.50 | 58.09 | 29.86 | 41.75 |
| SnapKV | 30.42 | 38.55 | 45.71 | 62.38 | 59.78 | 38.36 | 31.77 | 23.19 | 24.37 | 69.00 | 88.64 | 47.32 | 12.00 | 100.00 | 52.09 | 34.65 | 47.39 |
| CAKE | 30.07 | 41.66 | 45.96 | 64.45 | 59.80 | 36.18 | 31.64 | 22.70 | 24.23 | 68.00 | 89.11 | 47.06 | 10.50 | 100.00 | 52.56 | 35.24 | 47.48 |
| AdaKV | 29.58 | 38.02 | 45.63 | 61.46 | 57.33 | 37.91 | 31.43 | 22.40 | 24.13 | 71.00 | 88.86 | 46.64 | 11.50 | 100.00 | 52.54 | 33.95 | 47.02 |
| CriticalKV | 33.01 | 41.97 | 48.07 | 62.35 | 62.58 | 39.79 | 32.45 | 23.69 | 24.99 | 72.00 | 88.83 | 46.95 | 11.00 | 99.88 | 52.03 | 34.02 | 48.39 |
| DefensiveKV | 32.10 | 46.38 | 51.50 | 64.50 | 63.07 | 39.13 | 32.77 | 24.41 | 24.96 | 73.50 | 88.69 | 47.07 | 11.00 | 99.88 | 53.35 | 33.72 | 49.13 |
| Layer-DefensiveKV | 31.28 | 46.18 | 51.97 | 63.24 | 63.20 | 39.80 | 32.56 | 24.37 | 24.96 | 73.00 | 88.65 | 46.73 | 13.50 | 100.00 | 52.74 | 34.06 | 49.14 |

# N    DETAILS OF 16 DATASETS IN LONGBENCH

As a widely used long-context benchmark (Feng et al., 2024; Li et al., 2024; Zhang et al., 2024a), LongBench consists of 16 datasets across six task domains: single-document question answering (QA) (Kočiskỳ et al., 2018; Dasigi et al., 2021), multi-document QA (Yang et al., 2018; Ho et al., 2020; Trivedi et al., 2022), summarization (Huang et al., 2021; Zhong et al., 2021; Fabbri et al.,

Table 11: Detailed scores of 16 datasets on LongBench (80% cache size).

| Method | Single-Document QA | | | Multi-Document QA | | | Summarization | | | Few-shot Learning | | | Synthetic | | Code | | |
| | NrtvQA | Qasper | MF-en | Hotpot | 2WikiQA | Musique | GovRep | QMSum | MultiNews | TREC | TriviaQA | SAMSum | PCount | PR-en | Lcc | RB-P | Avg. |
|---|---|---|---|---|---|---|---|---|---|---|---|---|---|---|---|---|---|
| Llama-3.1-8B-Instruct, 80% Cache Size | | | | | | | | | | | | | | | | | |
| Full Cache | 29.55 | 44.68 | 55.82 | 57.59 | 48.89 | 32.61 | 34.40 | 25.51 | 26.83 | 73.00 | 92.36 | 43.27 | 7.38 | 99.50 | 63.44 | 52.36 | 49.20 |
| DuoAttention | 30.04 | 44.49 | 55.77 | 57.51 | 48.96 | 31.78 | 34.51 | 25.29 | 26.93 | 73.00 | 91.35 | 43.28 | 8.08 | 100.00 | 63.19 | 55.85 | 49.38 |
| StreamingLLM | 28.57 | 43.96 | 37.87 | 52.57 | 44.06 | 26.50 | 32.88 | 24.23 | 26.58 | 70.50 | 90.52 | 43.33 | 4.03 | 83.50 | 65.34 | 60.57 | 45.94 |
| SnapKV | 29.87 | 44.58 | 52.36 | 57.31 | 48.33 | 30.85 | 33.79 | 24.39 | 26.44 | 68.50 | 91.47 | 43.71 | 8.33 | 99.50 | 64.66 | 53.47 | 48.60 |
| CAKE | 29.53 | 43.76 | 56.26 | 57.28 | 47.81 | 30.71 | 33.26 | 25.14 | 26.59 | 72.50 | 92.75 | 42.82 | 9.60 | 99.50 | 63.75 | 51.87 | 48.95 |
| AdaKV | 29.93 | 44.89 | 57.17 | 56.55 | 48.34 | 32.59 | 34.13 | 25.14 | 26.36 | 73.00 | 91.80 | 43.54 | 8.66 | 99.50 | 64.12 | 53.32 | 49.31 |
| CriticalKV | 29.73 | 44.66 | 55.66 | 58.19 | 48.52 | 32.24 | 34.70 | 25.27 | 26.56 | 73.50 | 92.30 | 43.75 | 8.09 | 99.50 | 63.90 | 54.04 | 49.41 |
| DefensiveKV | 29.63 | 44.49 | 56.70 | 57.41 | 49.49 | 31.08 | 34.97 | 25.23 | 27.25 | 73.00 | 92.03 | 43.06 | 8.07 | 99.50 | 63.90 | 54.53 | 49.40 |
| Layer-DefensiveKV | 29.63 | 44.88 | 56.52 | 58.18 | 48.10 | 32.85 | 34.76 | 24.98 | 27.20 | 72.50 | 91.78 | 42.98 | 8.62 | 99.50 | 63.27 | 52.44 | 49.26 |
| Mistral-7B-Instruct-v0.3, 80% Cache Size | | | | | | | | | | | | | | | | | |
| Full Cache | 27.02 | 38.19 | 50.22 | 50.75 | 37.41 | 27.92 | 34.45 | 25.76 | 26.37 | 76.00 | 89.01 | 46.89 | 6.50 | 97.00 | 66.04 | 60.47 | 47.50 |
| DuoAttention | 26.07 | 36.33 | 50.03 | 51.37 | 36.30 | 26.79 | 33.95 | 25.90 | 26.61 | 76.00 | 88.91 | 47.11 | 4.50 | 97.50 | 65.53 | 60.53 | 47.09 |
| StreamingLLM | 23.78 | 35.71 | 38.09 | 50.73 | 37.79 | 24.82 | 33.51 | 24.31 | 25.85 | 50.00 | 83.38 | 42.98 | 2.65 | 82.00 | 34.22 | 45.51 | 39.71 |
| SnapKV | 26.42 | 36.01 | 49.00 | 50.06 | 36.40 | 28.56 | 34.08 | 24.54 | 25.89 | 73.50 | 88.91 | 46.86 | 6.00 | 96.00 | 66.28 | 60.67 | 46.82 |
| CAKE | 26.62 | 38.22 | 50.20 | 50.29 | 36.40 | 24.78 | 34.28 | 25.63 | 26.02 | 76.00 | 88.91 | 46.41 | 4.56 | 95.00 | 66.70 | 60.29 | 46.89 |
| AdaKV | 26.77 | 34.52 | 48.73 | 50.25 | 36.59 | 28.57 | 33.45 | 24.90 | 26.20 | 76.50 | 88.91 | 47.26 | 6.50 | 96.50 | 66.06 | 61.37 | 47.07 |
| CriticalKV | 27.34 | 36.72 | 49.04 | 51.26 | 36.94 | 27.13 | 33.85 | 25.32 | 25.88 | 76.50 | 88.91 | 47.28 | 6.05 | 97.50 | 65.78 | 59.56 | 47.19 |
| DefensiveKV | 27.79 | 38.29 | 50.34 | 50.86 | 37.84 | 27.63 | 34.24 | 25.87 | 26.25 | 75.50 | 89.21 | 47.29 | 6.00 | 97.00 | 66.14 | 60.95 | 47.58 |
| Layer-DefensiveKV | 27.06 | 38.23 | 51.56 | 50.76 | 36.48 | 28.32 | 34.54 | 25.37 | 26.44 | 76.00 | 89.04 | 47.21 | 5.50 | 98.00 | 66.10 | 61.13 | 47.61 |
| Qwen2.5-32B-Instruct, 80% Cache Size | | | | | | | | | | | | | | | | | |
| Full Cache | 30.88 | 46.13 | 52.87 | 63.59 | 59.75 | 38.78 | 32.59 | 24.35 | 24.95 | 72.00 | 88.26 | 47.05 | 12.50 | 100.00 | 49.64 | 34.24 | 48.60 |
| DuoAttention | N/A | N/A | N/A | N/A | N/A | N/A | N/A | N/A | N/A | N/A | N/A | N/A | N/A | N/A | N/A | N/A | N/A |
| StreamingLLM | 27.44 | 43.24 | 35.78 | 59.73 | 51.76 | 33.23 | 31.32 | 22.84 | 25.40 | 74.50 | 86.83 | 48.18 | 11.00 | 82.00 | 55.92 | 32.59 | 45.11 |
| SnapKV | 30.97 | 42.95 | 49.08 | 63.92 | 59.10 | 39.73 | 32.00 | 23.33 | 24.75 | 71.00 | 88.48 | 47.48 | 12.00 | 100.00 | 51.02 | 33.55 | 48.09 |
| CAKE | 30.32 | 45.68 | 52.23 | 63.96 | 61.16 | 39.91 | 32.17 | 23.67 | 25.06 | 70.50 | 88.77 | 47.39 | 11.00 | 100.00 | 51.02 | 34.84 | 48.60 |
| AdaKV | 30.42 | 42.62 | 48.92 | 62.98 | 60.45 | 39.19 | 31.77 | 23.22 | 24.82 | 71.00 | 88.69 | 46.87 | 10.50 | 100.00 | 50.18 | 33.60 | 47.83 |
| CriticalKV | 31.09 | 45.47 | 50.40 | 63.29 | 61.80 | 39.95 | 32.35 | 23.98 | 24.86 | 72.00 | 88.50 | 46.92 | 11.50 | 100.00 | 50.56 | 34.53 | 48.58 |
| DefensiveKV | 31.34 | 46.43 | 51.92 | 63.33 | 61.45 | 39.12 | 32.70 | 24.30 | 25.22 | 72.00 | 88.39 | 47.14 | 11.00 | 100.00 | 53.48 | 33.81 | 48.85 |
| Layer-DefensiveKV | 31.13 | 46.01 | 52.46 | 63.29 | 60.22 | 38.90 | 32.85 | 24.18 | 25.03 | 72.00 | 88.73 | 46.56 | 12.50 | 100.00 | 52.55 | 33.53 | 48.75 |

2019), few-shot learning (Joshi et al., 2017; Gliwa et al., 2019; Li & Roth, 2002), synthetic tasks (Bai et al., 2023), and code generation (Guo et al., 2023; Liu et al., 2023). The average token length across all 16 datasets is 6,711. Table 12 provides detailed information on the 16 datasets in LongBench.

Table 12: Details of 16 datasets in LongBench.

| Task | Task Type | Eval metric | Avg len | Language | Sample Num |
|---|---|---|---|---|---|
| NarrativeQA | Single-Doc. QA | F1 | 18,409 | EN | 200 |
| Qasper | Single-Doc. QA | F1 | 3,619 | EN | 200 |
| MultiFieldQA-en | Single-Doc. QA | F1 | 4,559 | EN | 150 |
| HotpotQA | Multi-Doc. QA | F1 | 9,151 | EN | 200 |
| 2WikiMultihopQA | Multi-Doc. QA | F1 | 4,887 | EN | 200 |
| MuSiQue | Multi-Doc. QA | F1 | 11,214 | EN | 200 |
| GovReport | Summarization | Rouge-L | 8,734 | EN | 200 |
| QMSum | Summarization | Rouge-L | 10,614 | EN | 200 |
| MultiNews | Summarization | Rouge-L | 2,113 | EN | 200 |
| TREC | Few-shot Learning | Accuracy | 5,177 | EN | 200 |
| TriviaQA | Few-shot Learning | F1 | 8,209 | EN | 200 |
| SAMSum | Few-shot Learning | Rouge-L | 6,258 | EN | 200 |
| PassageCount | Synthetic | Accuracy | 11,141 | EN | 200 |
| PassageRetrieval-en | Synthetic | Accuracy | 9,289 | EN | 200 |
| LCC | Code | Edit Sim | 1,235 | Python/C#/Java | 500 |
| RepoBench-P | Code | Edit Sim | 4,206 | Python/Java | 500 |

## O  LIMITATIONS

In this paper, we reveal for the first time the fragility of KV cache eviction and propose a defensive aggregation strategy for robust optimization. However, our work serves as a starting point and does not provide an in-depth investigation of broader robust optimization techniques. Future research can explore these techniques to further improve cache eviction performance.

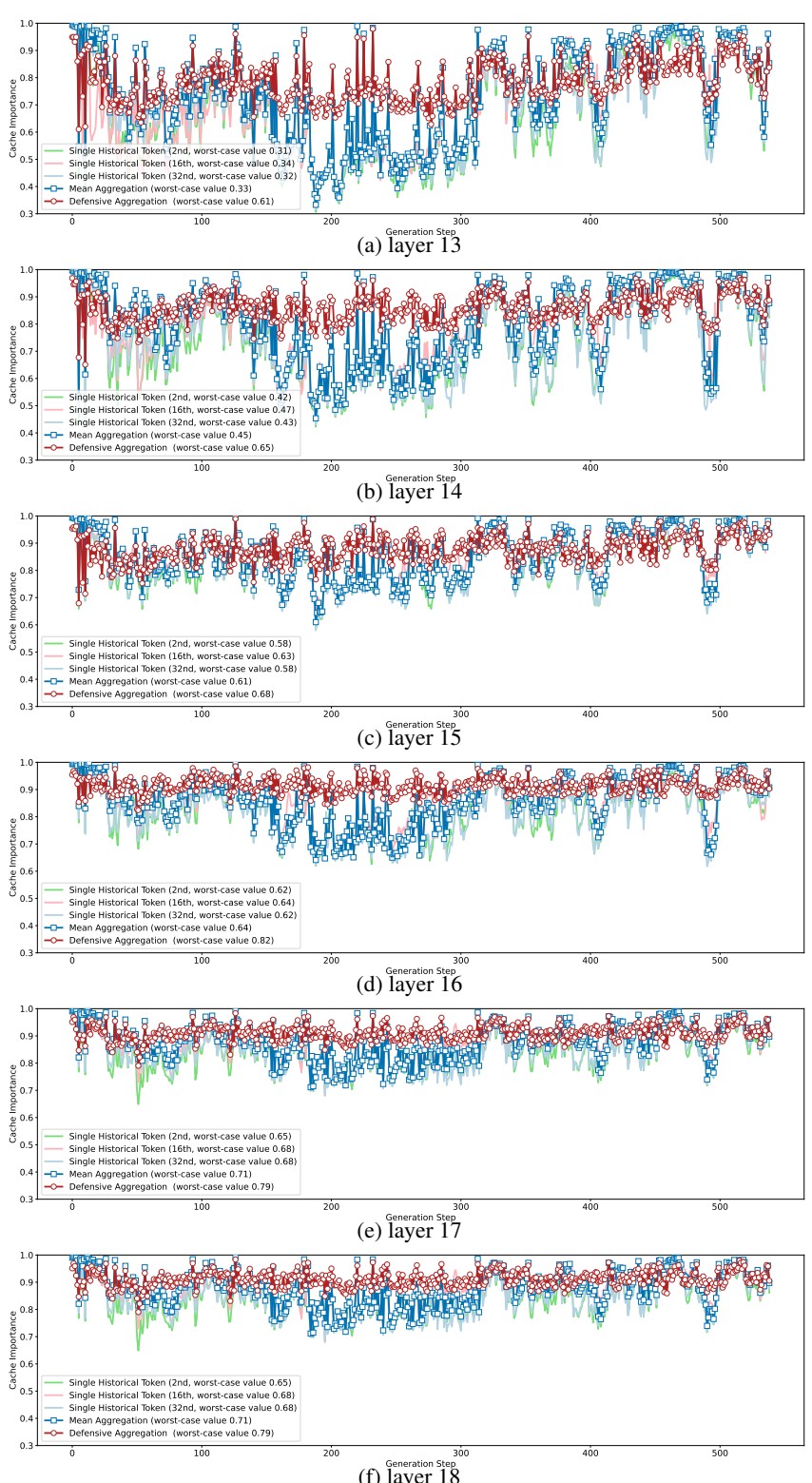

Figure 12: Visualization across different layers using Llama-3.1-8B with 50% cache size.

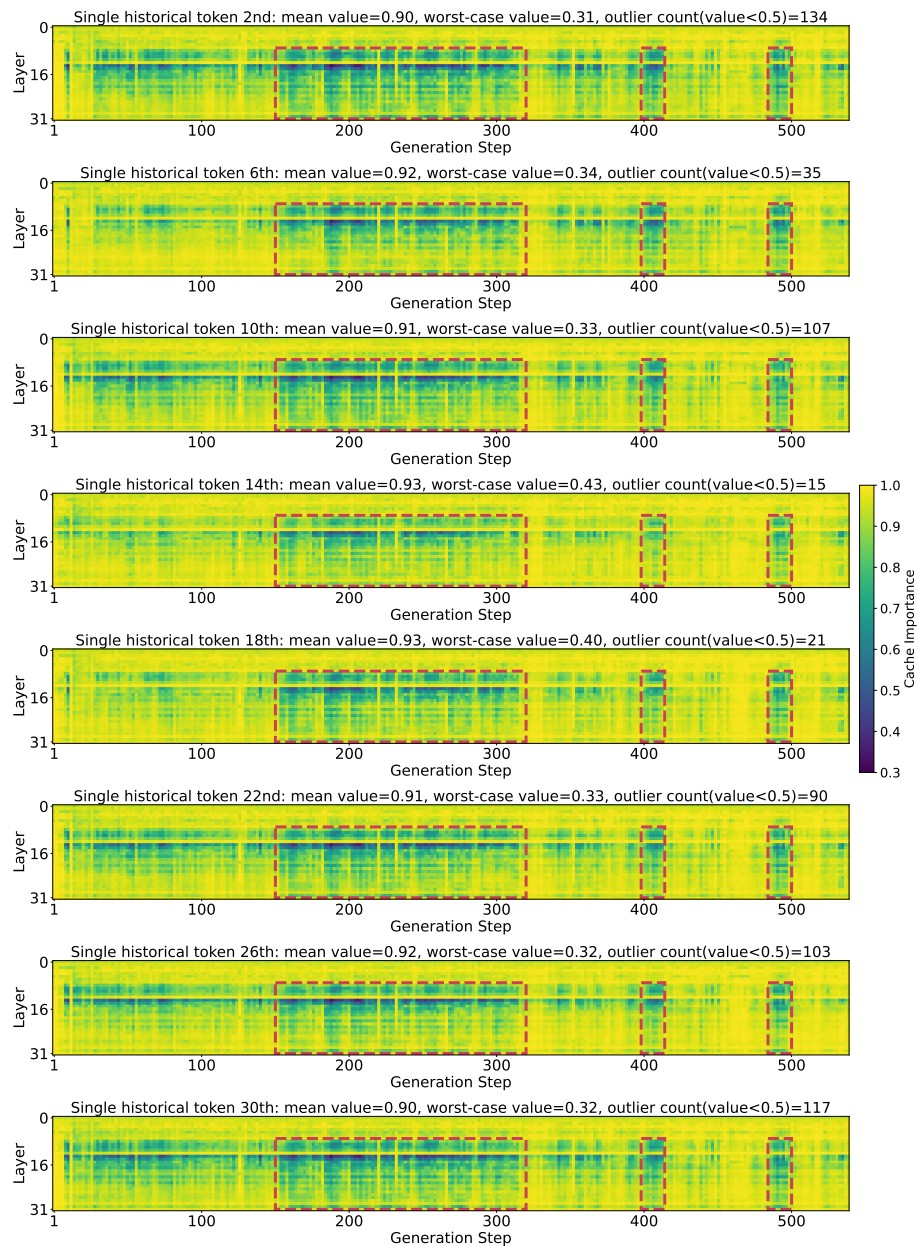

Figure 13: Breakdown of the stability assumption across different historical token measurements

Table 13: Single retrieval and multi retrieval templates in Needle-in-A-Haystack tests.

| | |
|---|---|
| Single retrieval | **Task Template:**
Some special magic numbers are hidden within the following text. Make sure to memorize it. I will quiz you about the numbers afterwards.
Paul Graham Essays.
...... One of the special magic numbers for {word} is: {number}. ......
What is the special magic number for {word} mentioned in the provided text?

The special magic number for {word} mentioned in the provided text is |
| Multi retrieval | **Task Template:**
Some special magic numbers are hidden within the following text. Make sure to memorize it. I will quiz you about the numbers afterwards.
Paul Graham Essays.
...... One of the special magic numbers for {word} is: {number-1}. ......
...... One of the special magic numbers for {word} is: {number-2}. ......
...... One of the special magic numbers for {word} is: {number-3}. ......
...... One of the special magic numbers for {word} is: {number-4}. ......
What are all the special magic numbers for {word} mentioned in the provided text?

The special magic numbers for {word} mentioned in the provided text are |

