# OpenReview forum: "DefensiveKV: Taming the Fragility of KV Cache Eviction in LLM Inference"
_ICLR.cc/2026/Conference — ICLR 2026 Poster_

### Official Review · Reviewer_Gsr3 · 2025-10-27

**Soundness:** 2
**Presentation:** 3
**Contribution:** 3
**Rating:** 4
**Confidence:** 5

**Summary:**

This work identifies a critical fragility in the common stability assumption underlying KV cache eviction methods for LLMs. It introduces a defensive aggregation strategy that manages worst-case risk by estimating peak importance and applying adaptive correction, moving beyond vulnerable mean aggregation. The resulting methods, DefensiveKV and Layer-DefensiveKV, achieve substantially lower generation quality loss under strong cache size constraints. Comprehensive evaluations across 18 datasets show these methods outperform prior art, reducing quality loss compared with baseline.

**Strengths:**

* S1) The concept of Defensive Aggregation presents a novel approach to KV cache eviction, and its demonstrated effectiveness is supported by empirical evidence across benchmarks including LongBench and NIAH.
* S2) The paper is written in a clear way, making it easy to follow the authors' ideas and understand their results.

**Weaknesses:**

* W1) The primary limitation of this work is its exclusive validation on dense and GQA-based model architectures. The proposed cache eviction method remains unverified for the increasingly prevalent Mixture-of-Experts (MoE) models. Since MoE architectures inherently reduce KV cache usage through selective parameter activation, the fundamental dynamics of token importance may differ. Consequently, the applicability and effectiveness of the defensive aggregation strategy in this critical and distinct setting are currently unknown.

* W2) The authors mention in their setup that "the context is compressed independently before the question is introduced" is beneficial for multi-turn dialogue, but no experimental validation was conducted on real multi-turn dialogue datasets.

* W3) While the paper demonstrates strong performance at cache retention rates of 20% and above, it does not systematically evaluate the proposed method under more extreme compression scenarios. The performance trajectory of DefensiveKV when cache size drops below 10% remains unclear. For example, HeadKV evaluates its performance when retaining an average of 128 to 1024 tokens on LongBench.

* W4) The authors' experiments primarily focus on long-text datasets. I think it would be better to test the performance of the proposed method on mathematical and reasoning tasks.

* W5) There is a lack of theoretical analysis or mathematical proof to demonstrate why defensive aggregation outperforms mean aggregation.

* W6) It would be helpful if the authors could provide additional information to support reproducibility.

**Questions:**

Please refer to Weaknesses.

---

> ### Author Response · Authors · 2025-11-18
> **Reply to Reviewer Gsr3 (Part 1)**
>
> Thank you for your dedicated effort during the review process. We sincerely appreciate you recognizing our work for identifying a critical fragility in a common assumption and proposing a novel approach that demonstrates strong empirical performance across wide benchmarks. Below, we address your remaining concerns in detail.
>
> ## Reply to Weakness1:
> > The primary limitation of this work is its exclusive validation on dense and GQA-based model architectures. ... Since MoE architectures inherently reduce KV cache usage through selective parameter activation, the fundamental dynamics of token importance may differ....
>
> **We would first like to clarify that MoE architectures only reduce the runtime activation parameters within the Feed-Forward Networks (FFN), and do not directly impact KV Cache usage in Attention Blocks.** Therefore, MoE does not reduce KV cache usage. Since the cache eviction methods operates within the attention blocks, it is agnostic to the modification of MoE in FFN structure. Therefore,  **our defensive aggregation strategy inherently generalize to MoE models without requiring any architecture-specific modifications.**
>
> The large overall size of MoE models demands substantial GPU memory for benchmarking, which is why no prior work evaluates these architectures.  **Following your advice, we have conducted additional experiments on the recently released Qwen-3-30B-A3B MOE model on Longbench. The results show that our methods provide substantial gains:  DefensiveKV and Layer-DefensiveKV incur only 5.6% and 3.8% quality loss on average, respectively, compared to 8.6% for the strongest baseline, CriticalKV.** These results provide strong evidence that our defensive aggregation strategy generalizes effectively to MoE models while maintaining its advantages.
>
> | Qwen3-30B-A3B, 20% Cache | NrtvQA | Qasper | MF-en | Hotpot | 2WikiQA | Musique | GovRep | QMSum | MultiNews | TREC | TriviaQA | SAMSum | PCount | PR-en | Lcc | RB-P | Avg. |
> |-|-|-|-|-|-|-|-|-|-|-|-|-|-|-|-|-|-|
> | Full Cache | 23.62 | 42.12 | 54.49 | 63.83 | 56.7 | 32.74 | 30.9 | 22.04 | 23.59 | 77 | 91.11 | 45.82 | 15.5 | 100 | 72.27 | 69.77 | 51.34 |
> | CriticalKV | 23.8 | 33.37 | 38.51 | 54.73 | 41.18 | 26.92 | **29.7** | 20.49 | 20.64 | 68 | **90.38** | 45.07 | 15 | 99.5 | 73.11 | 70.59 | 46.94 (↓8.6%)|
> | DefensiveKV | **24.19** | 36.66 | 42.75 | **59.14** | 44.91 | **31.95** | 29.56 | 21.13 | 20.94 | 69.5 | 90.18 | **45.58** | 15 | **100** | **73.16** | 70.55 | 48.45 (↓5.6%)|
> | Layer-DefensiveKV | 22.98 | **38.99** | **46.3** | 58.5 | **50.02** | 31.69 | 29.17 | **21.54** | **21.33** | **74** | 90.29 | 45.32 | **16** | **100** | 72.73 | **71.1** | **49.37** (↓3.8%)|
>
>
> ## Reply to Weakness2:
>
> > The authors mention in their setup that "the context is compressed independently before the question is introduced" is beneficial for multi-turn dialogue, but no experimental validation was conducted on real multi-turn dialogue datasets.
>
> Thank you for this question.
>
> **First, this setup has been widely adopted in related works and open-source project, like Ada-KV, CriticalKV, KVPRESS. We follow this setup primarily to evaluate our method in a more challenging and realistic scenario.** By compressing the context before any questions are known, we prevent the model from tailoring the compression to a specific query. In contrast, the alternative setting compresses the long text based on a specific question. The former setting is more challenging because the compressed cache must be robust enough to answer any future question, thereby more faithfully reflecting how KV cache compression would be applied in broader real-world scenarios. **This  naturally captures the multi-turn QA scenario, where multiple different questions are asked over the same compressed context.**
>
> **Second, we wish to clarify the distinction between "multi-turn QA" in our context and standard multi-turn dialogue datasets.** In the context of KV Cache compression, "multi-turn QA" involves a one-time compression of a given long text (e.g. document **often exceeding 10K tokens, and up to 32K in our evaluations**), which is then used to answer multiple, subsequent questions.  However, the regular multi-turn dialogue datasets capture conversational history between a user and a chatbot, but such datasets rarely feature long contexts where KV Cache compression would have significant benefits. This is why existing KV Cache compression methods have not been evaluated on these multi-turn dialogue datasets.
>
> **We hope this clarifies our rationale. However, if you believe a specific multi-turn dialogue dataset would be particularly informative for evaluation, please let us know. We would be happy to conduct additional targeted experiments.**

---

> > ### Author Response · Authors · 2025-11-18
> > **Reply to Reviewer Gsr3 (Part 2)**
> >
> > ## Reply to Weakness3:
> > > While the paper demonstrates strong performance at cache retention rates of 20% and above, it does not systematically evaluate the proposed method under more extreme compression scenarios. The performance trajectory of DefensiveKV when cache size drops below 10% remains unclear.
> >
> > Thank you for your suggestion. We have incorporated this discussion into the revised manuscript (see Appendix E).
> >
> > First, we would like to clarify that **a 20% cache already represents a extreme compression in above mentioned challenging setting.**  Our DefensiveKV yields a 3.9% average loss across three models, compared to 10.3% for the strongest baseline (CriticalKV) and around 20% for other baselines. In such cases, **the loss from baselines makes them impractical for real-world use, while our method keeps the degradation within a usable range.**
> >
> > Following your advice, **we have included additional results with 10% cache size, where our method continues to demonstrate a significant advantage over all baselines.** However, under such extreme compression ratios, all methods already suffer more than a 10%  loss, which implies limited practical value in real-world applications.
> >
> > **Please let us know if you would like results under other budgets—we will be happy to provide them.**
> >
> > |Llama-3.1-8B, 10% Cache| NrtvQA| Qasper | MF-en | Hotpot | 2WikiQA | Musique | GovRep | QMSum | MultiNews | TREC | TriviaQA | SAMSum | PCount | PR-en | Lcc | RB-P | Avg. |
> > |-|-|-|-|-|-|-|-|-|-|-|-|-|-|-|-|-|-|
> > | Full Cache | 29.55 | 44.68 | 55.82 | 57.59 | 48.89 | 32.61 | 34.4 | 25.51 | 26.83 | 73 | 92.36 | 43.27 | 7.38 | 99.5 | 63.44 | 52.36 | 49.2 |
> > | SnapKV | 23.25 | 20.77 | 23.54 | 43.11 | 22.29 | 18.57 | 24.13 | 19.51 | 20.43 | 43.00 | 92.01 | 43.18 | 7.35 | 51.50 | 64.80 | 54.17 | 35.73(↓27.4%) |
> > | CriticalKV(Strongest Baseline) | **27.62** | 25.02 | 28.78 | 43.49 | 23.50 | 20.67 | 25.69 | 21.39 | 21.10 | 48.50 | 92.17 | 42.84 | 7.30 | 63.50 | 66.61 | 54.09 | 38.27(↓22.2%) |
> > | DefensiveKV | 27.23 | 25.32 | 35.30 | 46.29 | 27.15 | 18.23 | 25.42 | 22.39 | 20.88 | **54.00** | 91.96 | **44.13** | 5.90 | **98.50** | 67.46 | 54.15 | 41.52(↓15.6%) |
> > | Layer-DefensiveKV | 27.30 | **28.53** | **38.09** | **49.12** | **30.80** | **22.44** | **25.76** | **22.42** | **21.41** | 52.00 | 92.01 | 43.94 | **8.68** | 98.00 | **67.66** | **56.01** | **42.76(↓13.1%)** |
> >
> >
> > ## Reply to Weakness 4：
> >
> > >The authors' experiments primarily focus on long-text datasets. I think it would be better to test the performance of the proposed method on mathematical and reasoning tasks.
> >
> > KV Cache Compression typically yields practical benefits with long-sequence inputs. Therefore, following prior works, we focused our evaluation on long-text benchmarks. **Additionally, following your suggestion, we evaluated our method on the popular mathematical dataset GSM8K.** Since cache compression methods require observing sufficient historical tokens, and the average input length in GSM8K is only 42.8 tokens, we filtered samples in both benchmarks to include only those with token lengths exceeding 64 for evaluation. On GSM8K, DefensiveKV and Layer-DefensiveKV demonstrated substantial improvements. For example, Layer-DefensiveKV achieved a score of 69.46 with a 60% cache, significantly surpassing the strongest baseline, CriticalKV, which scored 42.86.
> >
> >
> > | | 100% Cache | 90% Cache | 80% Cache | 70% Cache | 60% Cache |
> > |-|--|-|-|-|-|
> > | Strongest Baseline CriticalKV | 75.37| 71.43|66.01|54.68|42.86|
> > | DefensiveKV |75.37| 72.91| 67 | 62.07| 60.59|
> > | Layer-DefensiveKV|75.37 | **76.35** |**73.4**|**73.4**|**69.46**|

---

> ### Author Response · Authors · 2025-11-18
> **Reply to Reviewer Gsr3 (Part 3)**
>
> ## Reply to Weakness 5:
>
> > theoretical analysis or mathematical proof to demonstrate why defensive aggregation outperforms mean aggregation
>
> We agree that formal justification would strengthen our work beyond the empirical results. **Following your suggestion, we have added a new theoretical analysis section in the revised manuscript (Appendix C).** This section formally models the fragile stability phenomenon via a Bernoulli mixture model and mathematically proves that defensive aggregation is superior to mean aggregation for worst-case risk estimation.
>
> The key findings are summarized by the following theorems:
>
>
> 1.  **High Bias of Mean Aggregation (Theorem 1):** We prove that mean aggregation is a high-bias estimator of the true risk. Its expected value is mathematically anchored to the mean risk of the (low-risk) stable state, causing it to  severely underestimate the peak risk in fragile stability scenarios.
>
> 2.  **Lower Bias of Defensive Aggregation (Theorem 2):** We then prove that our worst-case risk estimator ($\tilde{R}_i = \max I_{j,i}$) offers significantly lower peak-risk estimation bias, with a formally proven expected bias strictly smaller than that of mean aggregation.
>
> 3.  **Robustness from Adaptive Correction (Corollary 1):** Finally, the adaptive correction step ($R_i = \max(\tilde{R}_i, \bar{R})$) provides a Bayesian-style safeguard. It addresses the "under-observation" failure mode—where all $m$ samples miss the rare spike. In this case, the algorithm discards the unreliable (low) observed value and substitutes the more robust head-level prior, ensuring a reliable lower bound.
>
> In summary, this new theoretical analysis demonstrates that our approach is a mathematically sound strategy that yields a lower-bias estimator for worst-case risk, specifically designed to overcome the critical failure modes of mean aggregation under fragile stability.
>
>
>
> ## Reply to Weakness 6：
>
> > It would be helpful if the authors could provide additional information to support reproducibility.
>
> We promise to open-source all code to ensure full reproducibility upon publication.
>
> Our implementation is based on a widely-used open-source cache compression framework from GitHub. We will submit a pull request to the official repository to facilitate easy evaluation of our algorithm by the community and to contribute to the broader advancement of the field.
>
>
> *****
>
> **If any concerns remain, please share them—we are happy to discuss further. Should our responses address your concerns, we would greatly appreciate you reconsidering our score. Thank you for your time and thoughtful feedback.**

---

> ### Comment · Reviewer_Gsr3 · 2025-11-22
>
> Thank you for the thoughtful response. I have two main concerns.
>
> * First, regarding MoE models, I agree that the description about MoE architectures inherently reducing KV cache usage is inaccurate. I mentioned this because I previously applied KV cache eviction methods such as AdaKV and HeadKV to DeepSeek-V2-Lite, using an average of 128 to 1024 KV pairs per cache. The model’s loss increased significantly compared to the dense model, by at least 20% on LongBench. While I am glad to see DefensiveKV perform well with 20% cache retention, I do not consider this an extreme compression ratio. This is related to my second point.
>
> * Second, HeadKV was evaluated in the range of 128 to 1024 KV pairs, and its paper reports that performance at 1024 KV on longBench even exceeds that of FullKV, which is consistent with my own findings. Although DefensiveKV can be combined with HeadKV, HeadKV already achieves nearly lossless performance at 10% cache retention on Longbench with dense models. Therefore, the benefit of testing DefensiveKV at 10% is unclear. It would be more meaningful to evaluate at 1% to 3% cache levels, where the combination of DefensiveKV and HeadKV may outperform methods like SnapKV or CriticalKV combined with HeadKV.
>
> I recognize that multi-turn dialogue scenarios, though common in practice, are seldom tested in research papers. Given the scope of this work, I will not emphasize this aspect.
>
> If the authors can adequately address these two points, I would be willing to raise my score. For now, I maintain my rating below the acceptance threshold

---

> > ### Author Response · Authors · 2025-11-23
> >
> > Thank you for your response; we appreciate the opportunity to discuss these points in depth.
> >
> > ## First, we would like to clarify a **key misunderstanding**: HeadKV **does not** achieve nearly lossless performance at 10% cache retention.
> >
> > As shown below, at 20% cache retention, HeadKV suffers a 13.3% quality drop. In contrast, our DefensiveKV and Layer-DefensiveKV limit the degradation to 5.1% and 2.4%, respectively.
> >
> > |LongBench, Llama-3.1-8B, 20% Cache|NrtvQA|Qasper|MF-en|Hotpot|2WikiQA|Musique|GovRep|QMSum|MultiNews|TREC|TriviaQA|SAMSum|PCount|PR-en|Lcc|RB-P|Avg.|
> > |-|-|-|-|-|-|-|-|-|-|-|-|-|-|-|-|-|-|
> > |Full Cache|29.55|44.68|55.82|57.59|48.89|32.61|34.40|25.51|26.83|73.00|92.36|43.27|7.38|99.50|63.44|52.36|49.20|
> > |HeadKV|27.60|32.14|34.86|49.69|31.35|23.86|28.28|22.67|22.52|53.50|91.86|43.01|8.14|90.50|**67.77**|54.5|42.64 (↓13.3%)|
> > |CriticalKV|29.81|32.58|34.96|52.34|36.24|26.37|28.35|23.52|23.24|56.50|90.80|43.37|**8.89**|93.00|67.05|54.99|43.88 (↓10.8%)|
> > |DefensiveKV|29.97|40.46|46.23|52.20|38.40|**28.06**|29.96|23.89|24.11|68.00|**91.58**|43.17|8.28|**100.00**|67.17|55.40|46.68 (↓5.1%)|
> > |Layer-DefensiveKV|**30.10**|**42.91**|**52.94**|**55.03**|**44.07**|27.00|**30.99**|**24.95**|**24.42**|**69.00**|91.30|**43.54**|8.38|**100.00**|67.60|**56.00**|**48.01 (↓2.4%)**|
> >
> > **While HeadKV paper claims that its performance at 1024 KV even exceeds FullCache, this is due to flaws in the earlier evaluation setting**, summarized as follows:
> >
> > 1. **Fixed cache size leads to evaluation bias due to imbalanced sample lengths in LongBench.** With a fixed cache of 1024, the actual retention rates differ greatly: 19.9% for Qasper, 38.9% for Multi_News, 32.3% for lcc, and only 8% for PR-en. In many cases, 1024 tokens retain far more than our 20% setting. This bias also affects single datasets. For example, in Multi_News, 15.5% of samples need no compression at all, and over 92% are shorter than 5K tokens, resulting in retaining higher than 20% cache budget. This skews results, since LLMs generally perform worse on longer contexts. If a model only works on short samples (≤1024) and fails on longer ones, all compression methods will appear same under a fixed 1024-token budget. These issues have been recognized by later works, which now use retention-ratio-based evaluations.
> >
> > 2. **Question-Aware Inflation.** Early works, including HeadKV and SnapKV, compress the concatenation of context and question, enabling the model to leverage the specific question for tailor compression, inflating its scores. However, this evaluation fail to reflect in real-world scenarios, where the context is compressed once and then used to answer unseen questions.  This issue—analyzed as “question-aware (query-postion dependent) compression”  in subsequent works, like AdaKV and DuoAttn. We instead follow the more rigorous protocol widely adopted by subsequent works: only the context is compressed, with questions provided afterward. The compressed context must support unseen questions, making the task considerably more challenging.
> >
> > |Longbench Dataset|Avg.Length|L< 1K Count(Ratio)|L<5K Count(Ratio)|L<10K Count(Ratio)|L>10K Count(Ratio)|
> > |-|-|-|-|-|-|
> > |Qasper (Single-Doc. QA)|5143|0(0.00%)|84(56.76%)|143(96.62%) |5(3.38%)|
> > |2WikiQA (Multi-Doc. QA)|7124|1(0.50%)|57(28.50%)|164(82.00%)|36(18.00%)|
> > |Multi_News (Summary)|2627|31(15.50%)|184(92.00%)|196(98.00%)|4(2.00%)|
> > |PR-en (Synthetic)|12305|0(0.00%)|0(0.00%)|6(3.00%)|194(97.00%)|
> > |lcc (Code)|3174|2(0.40%)|439(87.80%)|486(97.20%)|14(2.80%)|
> >
> > ## Secondly, regarding the MoE concern:
> >
> > As noted above, 20% retention represents extreme compression where all baselines **(including HeadKV) suffer >10% loss**, limiting practical applicability. Only **our method maintains <5% loss—an over 2x improvement.**
> >
> > Regarding your observation that AdaKV and HeadKV degrade on DeepSeek-V2-Lite, we attribute this to **model-specific characteristics,  not the MoE architecture.** For instance, on Qwen3-30B-A3B, both CriticalKV and DefensiveKV show consistent performance comparable to dense models.
> >
> > Moreover, **many works have observed that models differ in their inherent resilience to compression. For example, our Figure 6 shows, all methods perform better on Llama-3.1-8B than on Mistral-7B or Qwen2.5.** Investigating which model properties enable better KV compression is an interesting direction but beyond our current scope. We are happy to discuss this in the appendix to inspire further research.
> >
> > ***
> >
> > **We hope this clarifies your concerns and welcome further discussion. Please let us know if there are specific aspects you would like us to address. For instance, if experiments at extremely low retention rates (1%-3%) are necessary, we can provide them, though we expect all methods to suffer >30% quality loss, limiting practical value. We are preparing additional experiments to compare HeadKV and further demonstrate our  advantages. Due to the high cost of long-context evaluation, this will require additional time.**

---

> > > ### Comment · Reviewer_Gsr3 · 2025-11-23
> > >
> > > Thank you for your reply. It has addressed some of my concerns.
> > >
> > > I believe the evaluation setting is very important. Your point about a fixed cache size causing evaluation bias in LongBench due to varying sample lengths makes sense. However, in the context of KV cache eviction, the benefits are minimal for short texts. For example, with 1,000 tokens, even keeping only 20% saves just 800 tokens. But with 100,000 tokens, keeping 20% still leaves 20,000 tokens to handle. Therefore, the fixed cache size setting is also meaningful. Could you please report DefensiveKV's performance under a fixed cache size, averaged for sizes of 128 and 1024?

---

> > > > ### Author Response · Authors · 2025-11-25
> > > >
> > > > Thank you for your response. Following your suggestions, we have reported results for fixed cache sizes of 1024 and 128. We include both our DefensiveKV and CriticalKV methods, and further show the effectiveness of our Defensive Aggregation in improving HeadKV.
> > > >
> > > > **Under a fixed budget of 1024, our methods significantly reduce compression loss.** For instance, while CriticalKV results in a 24.7% performance drop, DefensiveKV lowers this loss to 10.7%. HeadKV incurs a 13.2% loss, which can be further reduced to just 6.1% when combined with Defensive Aggregation.
> > > >
> > > > | 1024 Budget | NrtvQA | Qasper | MF-en | Hotpot | 2WikiQA | Musique | GovRep | QMSum | MultiNews | TREC | TriviaQA | SAMSum | PCount | PR-en | Lcc | RB-P | Avg. |
> > > > |---|---|---|---|---|---|---|---|---|---|---|---|---|---|---|---|---|---|
> > > > | Full Cache | 29.55 | 44.68 | 55.82 | 57.59 | 48.89 | 32.61 | 34.40 | 25.51 | 26.83 | 73.00 | 92.36 | 43.27 | 7.38 | 99.50 | 63.44 | 52.36 | 49.20 |
> > > > | - | - | - | - | - | - | - | - | - | - | - | - | - | - | - | - | - | - |
> > > > | CriticalKV | 19.19 | 30.82 | 33.81 | 41.91 | 30.74 | 14.31 | 25.05 | 20 | 25.42 | 50.5 | 91.26 | 42.86 | 8.26 | 39 | 64.8 | 55.09 | 37.06(↓24.7%) |
> > > > | DefensiveKV | **22.14** | **42.77** | **47.38** | **44.52** | **33.55** | **15.88** | **27.36** | **21.9** | **26.31** | **62** | **92.4** | **43.18** | **6.54** | **96** | **65.76** | **55.58** | **43.95(↓10.7%)** |
> > > > | - | - | - | - | - | - | - | - | - | - | - | - | - | - | - | - | - | - |
> > > > | HeadKV | 22.58 | 40.74 | 40.21 | 46.34 | 40.08 | **20.27** | 28.13 | 21.56 | **26.16** | 54 | **91.21** | 43.03 | **6.66** | 81.5 | 66.08 | 55.16 | 42.73(↓13.2%) |
> > > > | HeadKV + Defensive Aggregation | **23.64** | **44.59** | **47.07** | **49.52** | **48.79** | 19.84 | **30.77** | **23.74** | 26.12 | **61.5** | 91.04 | **43.66** | 5.22 | **98.5** | **67.54** | **57.38** | **46.18(↓6.1%)** |
> > > >
> > > > **With an  extremely low budget of 128, our approach remains effective.** DefensiveKV reduces the performance loss from 44.2% (CriticalKV) to 43.8%, and applying Defensive Aggregation to HeadKV lowers the loss from 42.3% to 39.2%.  However, we respectfully note that such an extremely constrained budget may not fully reflect the practical utility of compression methods, as all approaches incur significant performance drops (~40%) that likely exceed the tolerance of real-world applications.
> > > >
> > > > | 128 Budget | NrtvQA | Qasper | MF-en | Hotpot | 2WikiQA | Musique | GovRep | QMSum | MultiNews | TREC | TriviaQA | SAMSum | PCount | PR-en | Lcc | RB-P | Avg. |
> > > > |---|---|---|---|---|---|---|---|---|---|---|---|---|---|---|---|---|---|
> > > > | --- | --- | --- | --- | --- | --- | --- | --- | --- | --- | --- | --- | --- | --- | --- | --- | --- | --- |
> > > > | Full Cache | 29.55 | 44.68 | 55.82 | 57.59 | 48.89 | 32.61 | 34.40 | 25.51 | 26.83 | 73.00 | 92.36 | 43.27 | 7.38 | 99.50 | 63.44 | 52.36 | 49.20 |
> > > > | - | - | - | - | - | - | - | - | - | - | - | - | - | - | - | - | - | - |
> > > > | CriticalKV | 13.23 | 13.18 | 18.14 | **29.78** | 14.14 | **9.26** | 18.93 | **17.45** | 19.71 | 27.5 | **91.8** | 38.21 | 1.03 | 5 | **63.83** | **57.94** | 27.45(↓44.2%) |
> > > > | DefensiveKV | **14.11** | **14.27** | **19.69** | 25.94 | **15.69** | 8.76 | **19.01** | 17.43 | **20.7** | **32.5** | 91.03 | **39.61** | **2.74** | **7.5** | 62.38 | 50.9 | **27.64(↓43.8%)** |
> > > > | - | - | - | - | - | - | - | - | - | - | - | - | - | - | - | - | - | - |
> > > > | HeadKV | 13.9 | **16.9** | 18.63 | 30.47 | 13.95 | 6.9 | 19.03 | 17.19 | 21.03 | 31 | 90.55 | 38.12 | 3.5 | 8 | 64.31 | **60.78** | 28.39(↓42.3%) |
> > > > | HeadKV + Defensive Aggregation| **15.8** | 16.47 | **20.49** | **31.07** | **17.16** | **9.2** | **21** | **17.5** | **22.32** | **36** | **90.73** | **38.33** | **3.53** | **15** | **66.29** | 57.85 | **29.92(↓39.2%)** |
> > > >
> > > > These results confirm the consistent effectiveness of our method. We will incorporate them into the revised manuscript and believe this addresses your concerns.

---

> ### Comment · Reviewer_Gsr3 · 2025-11-28
>
> Thank you for the reply. It has addressed my concerns. I will increase my score

---

### Official Review · Reviewer_dk2k · 2025-10-28

**Soundness:** 3
**Presentation:** 3
**Contribution:** 3
**Rating:** 6
**Confidence:** 3

**Summary:**

The paper “Taming the Fragility of KV Cache Eviction in LLM Inference” argues that existing KV cache eviction methods rely on a fragile stability assumption—that a fixed subset of cache entries stays important throughout generation. This makes the common mean aggregation approach unreliable under instability. The authors propose Defensive Aggregation, a linear-time method that manages worst-case risk instead of average importance. Building on this, they introduce DefensiveKV and Layer-DefensiveKV, which achieve over 2×–4× lower quality loss than state-of-the-art baselines (e.g., CriticalKV) at small cache budgets, with negligible computational overhead. The work pioneers a “worst-case risk-aware” perspective for robust and efficient LLM cache eviction.

**Strengths:**

1. strong experiments comparing with many previous baselines. improvement is significant.
2. New idea of adding online adapatation.
3. Clearly identifies and formalizes a previously overlooked fragility in the “stability assumption” behind KV cache eviction and proposes a simple, elegant, and computationally cheap fix (Defensive Aggregation).

**Weaknesses:**

1. The “worst-case” heuristic lacks formal guarantees or ablation beyond empirical trends. Need to show some results on why this worst case is guaranteed besides it works empirically.
2. No discussion on interaction with quantization, retrieval, or offloading beyond brief appendix notes.

**Questions:**

Can we have some more formal guarantee or experiments to show the worst case scenario is prevented?
Can we give some discussion on how this online process can be potentially integrated into inference?

---

> ### Author Response · Authors · 2025-11-18
> **Reply to Reviewer dk2k**
>
> Thank you for your dedicated effort during the review process. We appreciate your recognition of our strong experimental results and novel contributions, particularly our simple yet effective solution to the previously overlooked fragility in KV cache eviction. We address your remaining concerns below.
>
> ## Reply to Weakness1 & Question1:
> > The “worst-case” heuristic lacks formal guarantees or ablation beyond empirical trends.
> > Can we have some more formal guarantee or experiments to show the worst case scenario is prevented?
>
> We agree that formal justification would strengthen our work beyond the empirical results. **Following your suggestion, we have added a new theoretical analysis section in the revised manuscript (Appendix C).** This section formally models the fragile stability phenomenon via a Bernoulli mixture model and mathematically proves that defensive aggregation is superior to mean aggregation for worst-case risk estimation.
>
> The key findings are summarized by the following theorems:
>
> 1.  **High Bias of Mean Aggregation (Theorem 1):** We prove that mean aggregation is a high-bias estimator of the true risk. Its expected value is mathematically anchored to the mean risk of the (low-risk) stable state, causing it to  severely underestimate the peak risk in fragile stability scenarios.
>
> 2.  **Lower Bias of Defensive Aggregation (Theorem 2):** We then prove that our worst-case risk estimator ($\tilde{R}_i = \max I_{j,i}$) offers significantly lower peak-risk estimation bias, with a formally proven expected bias strictly smaller than that of mean aggregation.
>
> 3.  **Robustness from Adaptive Correction (Corollary 1):** Finally, the adaptive correction step ($R_i = \max(\tilde{R}_i, \bar{R})$) provides a Bayesian-style safeguard. It addresses the "under-observation" failure mode—where all $m$ samples miss the rare spike. In this case, the algorithm discards the unreliable (low) observed value and substitutes the more robust head-level prior, ensuring a reliable lower bound.
>
> In summary, this new theoretical analysis demonstrates that our approach is a mathematically sound strategy that yields a lower-bias estimator for worst-case risk, specifically designed to overcome the critical failure modes of mean aggregation under fragile stability.
>
> ## Reply to  Weakness2 & Question2:
> > No discussion on interaction with quantization, retrieval, or offloading beyond brief appendix notes.
> > Can we give some discussion on how this online process can be potentially integrated into inference?
>
> Both existing KV cache eviction methods, including our enhanced method, DefensiveKV, can be seamlessly integrated with orthogonal efficient inference techniques. For example:
>
> * KV Cache Quantization: Apply KV eviction to compress the cache first, then quantize the reduced cache to further minimize memory footprint.
> * KV Cache Retrieval: First compress the cache via KV eviction, then perform retrieval operations only on the compressed cache. This reduces both retrieval overhead and overall memory usage.
> * KV Cache Offloading: Applying KV eviction to compress the cache before offloading reduces both storage requirements and data transfer overhead. This, in turn, enables more extensive and efficient caching in slower memory systems.
>
>
> As a case study, we demonstrate integration with KV cache quantization. Specifically, we use DefensiveKV to keep only 40% of cache entries, then apply 4-bit quantization. **This yields 10% of the original memory usage with less than a 1-point drop in generation quality.** This is not achievable with quantization alone under the same memory budget, since it would require approximately 1.6-bit quantization. It also substantially outperforms using cache eviction alone to reduce the cache size to 10%. Thus, combining these techniques is a highly effective solution in real-world deployment. **We have added this discussion in the revised manuscript (Appendix H).**
>
> |Llama-3.1-8B|NrtvQA|Qasper|MF-en|Hotpot|2WikiQA|Musique|GovRep|QMSum|MultiNews|TREC|TriviaQA|SAMSum|PCount|PR-en|Lcc|RB-P|Avg.|
> |-|-|-|-|-|-|-|-|-|-|-|-|-|-|-|-|-|-|
> |Full Cache (100% memory)|29.55|44.68|55.82|57.59|48.89|32.61|34.4|25.51|26.83|73|92.36|43.27|7.38|99.5|63.44|52.36|49.2|
> |DefensiveKV-40% Cache (40% memory)|30.07|46.37|54.9|57.5|45.97|28.85|33.7|24.69|26.2|71.5|91.78|43.69|9.88|100|66.25|55.97|49.21|
> |-|-|-|-|-|-|-|-|-|-|-|-|-|-|-|-|-|-|
> |DefensiveKV-10% Cache (10% memory)|27.23|25.32|35.30|46.29|27.15|18.23|25.42|22.39|20.88|54.00|**91.96**|**44.13**|5.90|98.50|**67.46**|54.15|41.52|
> |DefensiveKV-40% Cache-int4 (10% memory)|**30.63**|**44.62**|**54.44**|**56.14**|**42.9**|**28.15**|**33.79**|**25.15**|**25.92**|**70.5**|91.28|43.76|**7.55**|**100**|65.71|**56.23**|**48.55**|
>
> *****
>
> **If any concerns remain, please share them—we are happy to discuss further. Should our responses address your concerns, we would greatly appreciate you reconsidering our score. Thank you for your time and thoughtful feedback.**

---

### Official Review · Reviewer_WZVm · 2025-10-30

**Soundness:** 3
**Presentation:** 3
**Contribution:** 3
**Rating:** 6
**Confidence:** 3

**Summary:**

This paper first argues that most of the existing methods (such as snapkv, criticalkv) are committed to improving the scoring step (i.e., looking for better importance indicators). They use simple average aggregation by default because they rely on a stability assumption: a small group of key cache entries will remain important throughout the generation process. However, the importance of cache entries may suddenly decline sharply at some time, resulting in extreme cases or outliers. The paper shows that the worst-case risk should be addressed. To do this,  this paper proposes a new defensive aggregation strategy to control the worst-case risk, which combines worst-case risk estimation and adaptive prior-risk correction. A DefensiveKV, which is a basic version, and a Layer-DefensiveKV are proposed, which integrate a layer-wise budget allocation strategy, respectively.

**Strengths:**

1. This new aggregation strategy has the same linear time complexity as the average aggregation strategy, and the additional computational overhead is negligible.

2. Significant performance improvement

3. Good writing and illustrations.

**Weaknesses:**

1. Is it possible that Adaptive Prior-Risk Correction will lead to: In a head with high risk, it may wrongly raise the score of really unimportant items; In a head with generally low risk, it cannot discover those rare key items that do not show high risk in the limited observation window.


2. It seems that in the code dataset, most SOTA methods provide elegant results. Could you provide some discussion about this?

3. In cirticalKV, the score is evaluated by using the attention weights and norm of V*W^O, which shows an excellent performance. This paper also adopts the same strategy. I also want to see some discussion about this: How many output differences does this paper's method reduce?

4. Algorithm 1 is conducted in layer-wise or head-wise?

**Questions:**

Please see Weaknesses.

---

> ### Author Response · Authors · 2025-11-18
> **Reply to Reviewer WZVm**
>
> We are grateful for your time and effort in reviewing our paper. We sincerely appreciate your positive assessment of our work, particularly your recognition that our new aggregation strategy achieves significant performance improvements with negligible computational overhead.  We address your remaining concerns below.
>
> ## Reply to Weakness1:
> > Is it possible that Adaptive Prior-Risk Correction will lead to...
>
>
> We appreciate you raising this important point. While the trade-off you described is a valid consideration, we believe the Adaptive Prior-Risk Correction effectively navigates it for two reasons:
>
> **First, the extensive experiments provide strong evidence for the robustness and effectiveness of our method.** We evaluated our algorithm across three different families of LLMs with varying sizes. The evaluations were conducted on a comprehensive suite of 18 datasets spanning seven distinct task domains. The consistent and significant performance gains observed in these diverse settings demonstrate the practical reliability of our methods.
>
> **Second, the design of the Adaptive Prior-Risk Correction mechanism inherently mitigates these potential failure modes.** The core of the correction is the formula:
>
>
> $R_i = \max(\tilde{R}_i, \bar{R}) $
>
>
> The threshold $\bar{R}$ is simply the mean observed risk of all cache entries in the current head. When using the mean of a batch as the clipping threshold, a certain proportion of data points will always be above the mean and remain unaffected, while the others will be clipped. This design ensures that neither all values are clipped nor all remain unchanged. This hyperparameter-free mechanism inherently stabilizes the process and prevents extreme cases.
>
>
> ## Reply to Weakness2:
> > It seems that in the code dataset, most SOTA methods provide elegant results. Could you provide some discussion about this?
>
> The quality loss of KV cache compression technique is influenced by several factors: the compression method, the choice of LLMs, the cache budget, and, critically, the complexity of the task. **Our findings are consistent with prior work (e.g., CriticalKV, AdaKV, SnapKV), which also suggest that code-related tasks often exhibit a lower sensitivity to cache compression.** This is because code generation and understanding frequently rely on local, recent context, with less dependence on long-range information.
>
> In contrast, tasks that demand a more complex and frequent reliance on long-range context—such as Single-Document QA, Multi-Document QA, and Summarization—present a greater challenge for existing compression methods, leading to more significant performance degradation. It is precisely in these more demanding scenarios that our proposed method demonstrates a more pronounced advantage and offers substantial improvements.
>
>
> ## Reply to Weakness3:
> > In cirticalKV, the score is evaluated by using the attention weights and norm of V*W^O, which shows an excellent performance. This paper also adopts the same strategy. I also want to see some discussion about this: How many output differences does this paper's method reduce?
>
> In DefensiveKV, we adopt the same scoring strategy as the strongest baseline, CriticalKV, but modify the aggregation step. This modification makes the method more robust when the stability assumption fails during future decoding, thereby further reducing the output differences.
>
> To validate this, we record the output differences between the last-layer hidden states of each method and those of the full-cache setting on the LongBench QA datasets under 20% cache sizes. As shown below, the differences increase with the decoding step, because the generated tokens gradually diverge from those under the full cache.
>
> Overall, **our DefensiveKV consistently yields lower differences than CriticalKV.** For example, the average \(L_2\) difference for DefensiveKV is 69.18, compared to 98.74 for CriticalKV, a reduction of about 30%. This provides clear empirical evidence that the worst-case risk control strategy in DefensiveKV substantially reduces subsequent output differences.
>
> |L2 difference|DefensiveKV vs. Full Cache|CriticalKV vs. Full Cache|
> |-|-|-|
> |Step 1|**44.16**|61.65|
> |Step 2|**65.46**|96.51|
> |Step 3|**71.35**|108.04|
> |Step 4|**80.50**|113.36|
> |Step 5|**84.45**|114.13|
> |Average|**69.18**|98.74|
>
>
>
> ## Weakness4: Algorithm 1 is conducted in layer-wise or head-wise?
>
> Algorithm 1 operates in a head-wise manner. This head-wise approach forms the basis of Adaptive Prior-Risk Correction, where we use head-specific prior risk for risk estimation and correction.
>
>
> *****
>
> **If any concerns remain, please share them—we are happy to discuss further. Should our responses address your concerns, we would greatly appreciate you reconsidering our score. Thank you for your time and thoughtful feedback.**

---

> > ### Comment · Reviewer_WZVm · 2025-11-22
> >
> > This is a good rebuttal.
> >
> > However, I am also confused about the L2 difference. Are these L2 differences evaluated on the V*W^O?

---

> > > ### Author Response · Authors · 2025-11-23
> > >
> > > Thank you for your response.
> > >
> > > Both L2 difference of our method and CriticalKV are evaluated using the V*W^O scoring, which shows that our approach can further reduce  the output differences on top of CriticalKV.
> > >
> > > Furthermore, our defensive aggregation is orthogonal to improvements in scoring. In our paper, we combined defensive aggregation with the SOTA scoring baseline, CriticalKV, to form DefensiveKV. If future methods achieve better scoring and further reduce perturbations, we believe our defensive aggregation can also enhance those approaches.

---

> > > > ### Comment · Reviewer_WZVm · 2025-11-23
> > > >
> > > > Thank you very much. As we all know, CriticalKV is specifically designed for minimizing the output difference. The only goal of  CriticalKV is to minimize output difference. Why is this method better than CriticalKV?

---

> > > > > ### Author Response · Authors · 2025-11-24
> > > > >
> > > > > Thank you for your insightful question!
> > > > >
> > > > > The fundamental reason for our improvement is that our work targets **a previously unrecognized vulnerability: the fragility of the underlying stability assumption**.
> > > > >
> > > > > **Limitations of CriticalKV:** CriticalKV is designed to reduce output differences **built on the assumption of underlying stability**. It introduces a novel scoring strategy that incorporates $VW^O$ alongside attention weights.  Although this reduces perturbation better than "attention-weights-only" methods, **both approaches neglect the fragility of stability assumption during generation**. Consequently, the actual perturbation in CriticalKV during generation is still constrained by the fragility of this underlying assumption.
> > > > >
> > > > > **Our Contribution:** We identify the **fragility of underlying stability assumption** as a critical oversight in prior work. To address this new perspective, we propose "Defensive Aggregation," which mitigates the worst-case risks caused by instability. Thus, our method effectively reduces actual real-world perturbation and significantly lowers the loss induced by compression.
> > > > >
> > > > > In essence, while CriticalKV minimizes perturbation **within the bounds of the stability assumption**, our work addresses **the real-world fragility of that foundational assumption**, resulting in superior performance in practical scenarios.
> > > > >
> > > > > ***
> > > > >
> > > > >
> > > > > We hope our response has clarified all your concerns, and we would sincerely appreciate your consideration of our rebuttal when updating your evaluation.

---

### Official Review · Reviewer_CK4Z · 2025-10-31

**Soundness:** 3
**Presentation:** 3
**Contribution:** 2
**Rating:** 6
**Confidence:** 3

**Summary:**

This paper revisits the standard scoring–aggregation pipeline for KV cache eviction in LLM inference and argues that the usual mean aggregation is fragile because the importance of cache entries can vary abruptly during generation. Based on this, this paper introduces Defensive Aggregation, a linear-time, two-step rule that first estimates a worst-case risk per entry via a maximum over historical-token scores and then applies an adaptive prior-risk correction, yielding two methods, DefensiveKV and Layer-DefensiveKV with layer-wise budget allocation. Across LongBench and Needle-in-a-Haystack, the introduced methods achieve better performance versus baselines at various cache budgets.

**Strengths:**

* The introduced method is well-motivated by convincing observations about the fragility of importance stability.

* Thorough experiments and analyses are presented in the paper.

**Weaknesses:**

* Could you elaborate on the rationale behind fixing the historical window size at 32 tokens? Are there any experiments to evaluate the sensitivity to this choice?

* The cache budget settings could be more varied. 20% cache is not a strictly small budget. Would it be possible to provide experimental results under small budgets such as 1% and 2% to demonstrate how the method behaves under extreme compression conditions?

* Missing discussion on non-scoring-aggregation pipelines in the paper such as “Lacache: ladder-shaped kv caching for efficient long-context modeling of large language models (ICML2025)," which would help situate DefensiveKV more clearly within the broader space of KV cache compression methods.

* Why on needle-in-a-haystack tasks, DefensiveKV outperforms Layer-DefensiveKV sometimes? Could you please provide any insight into under what circumstances DefensiveKV/Layer-DefensiveKV will perform better? A short discussion on this might help readers better understand their trade-offs.

**Questions:**

Will the code be open-sourced? I think it would be valuable for the community.

---

> ### Author Response · Authors · 2025-11-18
> **Reply to Reviewer CK4Z (Part 1)**
>
> Thank you for your dedicated effort during the review process. We sincerely appreciate your recognition of our work’s convincing observations， clear motivation, thorough experiments, and analyses. Below, we address your remaining concerns in detail.
>
>
>
> ## Reply to Weakness 1
> > the rationale behind fixing the historical window size at 32 tokens
>
>
> Our method follows the default setting of the strongest baseline, CriticalKV, which has a default window size of 32. Following your advice, we conducted an ablation study with window sizes of 16 and 64. The results show that DefensiveKV achieves average scores ranging from 46.02 to 46.68, while Layer-DefensiveKV scores between 47.53 and 48.01. Both substantially outperform the strongest baseline, CriticalKV (43.88).
>
> We have added this discussion to the Section 4.4 of the revised manuscript.
>
>
>
>
> | Llama-3.1-8B, 20% Cache | window |  NrtvQA | Qasper | MF-en | Hotpot | 2WikiQA | Musique | GovRep | QMSum | MultiNews | TREC | TriviaQA | SAMSum | PCount | PR-en | Lcc | RB-P | Avg. |
> |-|-|-|-|-|-|-|-|-|-|-|-|-|-|-|-|-|-|-|
> | Full Cache | N/A | 29.55 | 44.68 | 55.82 | 57.59 | 48.89 | 32.61 | 34.40 | 25.51 | 26.83 | 73.00 | 92.36 | 43.27 | 7 | 99.50 | 63.44 | 52.36 | 49.20 |
> | CriticalKV | 32 | 29.81 | 32.58 | 34.96 | 52.34 | 36.24 | 26.37 | 28.35 | 23.52 | 23.24 | 57 | 90.80 | 43.37 | 8.89 | 93.00 | 67.05 | 54.99 | 43.88 |
> |-|-|-|-|-|-|-|-|-|-|-|-|-|-|-|-|-|-|-|
> | DefensiveKV | 16 | 29.75 | 35.43 | **46.54** | **55.51** | 36.41 | 25.64 | 29.20 | 23.68 | 23.34 | 66 | **91.69** | **44.02** | 7.31 | **100.00** | 67 | 55.10 | 46.02 |
> | DefensiveKV | 32 | **29.97** | **40.46** | 46.23 | 52.20 | **38.40** | **28.06** | **29.96** | **23.89** | **24.11** | **68** | 91.58 | 43.17 | **8.28** | **100.00** | **67.17** | **55.40** | **46.68** |
> | DefensiveKV | 64 | 28.98 | 36.09 | 45.49 | 56.69 | 39.31 | 27.50 | **30.63** | 24.12 | 23.31 | **68** | 91.13 | 43.35 | 6.77 | **100.00** | 66.72 | 55.04 | 46.41 |
> |-|-|-|-|-|-|-|-|-|-|-|-|-|-|-|-|-|-|-|
> | Layer-DefensiveKV | 16 | 29.64 | 41.23 | 49.21 | 54.59 | 41.31 | **29.05** | 30.19 | 24.22 | 24.25 | 68.50 | **91.86** | **44.04** | **9** | 99.50 | **68.82** | 55.00 | 47.53 |
> | Layer-DefensiveKV | 32 | 30.10 | 42.91 | **52.94** | 55.03 | **44.07** | 27.00 | 30.99 | **24.95** | **24.42** | 69 | 91.30 | 43.54 | 8.38 | **100.00** | 67.60 | 56.00 | **48.01** |
> | Layer-DefensiveKV | 64 | **30.51** | **43.40** | 51.42 | **55.36** | 40.03 | 26.23 | **31.60** | 24.70 | 24.29 | **70** | 91.61 | 43.21 | 6.92 | **100.00** | 68.69 | **56.46** | 47.75 |
>
> ## Reply to Weakness2
> > 20% cache is not a strictly small budget. Would it be possible to provide experimental results under small budgets to demonstrate behaves under extreme compression conditions?
>
>
> Thank you for your suggestion. We have incorporated this discussion into the revised manuscript (see Appendix D).
>
> In fact, **a 20% cache already represents a extreme compression in above mentioned challenging setting.** Our DefensiveKV yields a 3.9% average loss across three models, compared to 10.3% for the strongest baseline (CriticalKV) and around 20% for other baselines. In such cases, the loss from baselines makes them impractical for real-world use, while our method keeps the degradation within a usable range.
>
> **Following your advice, we have included additional results with 10% cache size, where our method continues to demonstrate a significant advantage over all baselines.**  However, when the cache ratio is further reduced to 2% or even 1%, the resulting performance degradation will becomes even more unacceptable, making impractical for real-world applications.
>
> **Please let us know if you would like results under other budgets—we will be happy to provide them.**
>
> | Llama-3.1-8B, 10% Cache | NrtvQA| Qasper | MF-en | Hotpot | 2WikiQA | Musique | GovRep | QMSum | MultiNews | TREC | TriviaQA | SAMSum | PCount | PR-en | Lcc | RB-P | Avg. |
> |-|--|-|-|-|-|--|-|-|-|--|-|-|--|-|-|--|--|
> | Full Cache | 29.55 | 44.68 | 55.82 | 57.59 | 48.89 | 32.61 | 34.4 | 25.51 | 26.83 | 73 | 92.36 | 43.27 | 7.38 | 99.5 | 63.44 | 52.36 | 49.2 |
> | SnapKV | 23.25 | 20.77 | 23.54 | 43.11 | 22.29 | 18.57 | 24.13 | 19.51 | 20.43 | 43.00 | 92.01 | 43.18 | 7.35 | 51.50 | 64.80 | 54.17 | 35.73(↓27.4%) |
> | AdaKV | 24.54 | 22.69 | 25.47 | 41.86 | 22.59 | 16.74 | 24.52 | 20.17 | 20.81 | 46.00 | 92.25 | 43.36 | 7.23 | 50.50 | 66.25 | 54.84 | 36.24(↓26.3%) |
> | CriticalKV | **27.62** | 25.02 | 28.78 | 43.49 | 23.50 | 20.67 | 25.69 | 21.39 | 21.10 | 48.50 | 92.17 | 42.84 | 7.30 | 63.50 | 66.61 | 54.09 | 38.27(↓22.2%) |
> | DefensiveKV | 27.23 | 25.32 | 35.30 | 46.29 | 27.15 | 18.23 | 25.42 | 22.39 | 20.88 | **54.00** | 91.96 | **44.13** | 5.90 | **98.50** | 67.46 | 54.15 | 41.52(↓15.6%) |
> | Layer-DefensiveKV | 27.30 | **28.53** | **38.09** | **49.12** | **30.80** | **22.44** | **25.76** | **22.42** | **21.41** | 52.00 | 92.01 | 43.94 | **8.68** | 98.00 | **67.66** | **56.01** | **42.76(↓13.1%)** |

---

> ### Author Response · Authors · 2025-11-18
> **Reply to Reviewer CK4Z (Part 2)**
>
> ## Reply to Weakness 3:
> > Missing discussion on non-scoring-aggregation pipelines in the paper such as “Lacache: ladder-shaped kv caching for efficient long-context modeling of large language models (ICML2025)," which would help situate DefensiveKV more clearly within the broader space of KV cache compression methods.
>
>
> Thank you for highlighting this work. **We have added a discussion of Lacache and non-scoring-aggregation pipelines in the Related Work section of the revised manuscript.** The main points are summarized as follows:
>
> DefensiveKV enhances mainstream scoring-aggregation pipelines, which  focus on compression quality. In contrast, LaCache and StreamingLLM employ a fixed-pattern paradigm, prioritizing operational efficiency. These approaches suit different scenarios.
>
> Typically, compression is infrequent (e.g., after prefilling or even offline for prefixed caching), so quality is crucial and compression speed less so. In these cases, the accuracy loss of fixed-pattern methods becomes a major limitation: as shown in Figure 2 of our manuscript, StreamingLLM incurs over 40% quality loss, while the SOTA scoring-aggregation baseline sees 10.3%. Our DefensiveKV further reduces this to just 3.9%. Lacache, though somewhat better than StreamingLLM, still falls short of scoring-aggregation methods. This gap is why recent research mainly focuses on the scoring-aggregation framework. However, fixed-pattern methods are advantageous when compression operation is performed frequently (e.g. every decoding step), where their operational efficiency becomes the core advantage.
>
> Overall, our work provides a simple, effective enhancement for the dominant  scoring-aggregation methods, which is better suited for common, less frequent compression scenarios. While we acknowledge the utility of fixed-pattern methods in specific contexts, our method is not designed to enhance them. We have clarified this in the revised paper.
>
> ## Reply to Weakness 4:
>
> > Why on needle-in-a-haystack tasks, DefensiveKV outperforms Layer-DefensiveKV sometimes? Could you please provide any insight into under what circumstances DefensiveKV/Layer-DefensiveKV will perform better? A short discussion on this might help readers better understand their trade-offs.
>
>
> Thank you for the question.
>
> Based on our DefensiveKV, Layer-DefensiveKV further incorporates an inter-layer scheduling strategy studies in related works for a more comprehensive evaluation. Its main advantage is the better performance under tight cache budgets on challenging tasks—a strength that may not be evident in simpler tests.
>
>
> For example on complex benchmarks like LongBench， while both DefensiveKV and Layer-DefensiveKV are effective with larger caches (40-80%), Layer-DefensiveKV excels under aggressive compression. At a 20% budget, it cuts the Llama-3.1-8B performance loss from 5.1% (DefensiveKV) to 2.3%—a trend also validated across Mistral and Qwen models. Conversely, "needle-in-a-haystack" is a simpler task where both methods achieve near-lossless results, with minor score variations (~1 point) that are not indicative of meaningful compression loss. However, under extreme conditions (Mistral-7B-v0.3 at 10% cache), Layer-DefensiveKV (161) still significantly outperforms DefensiveKV (139).
>
> In summary, Layer-DefensiveKV consistently excels in more demanding, budget-constrained settings. On simpler tasks where both methods operate with minimal information loss, their performance is nearly identical to the full-cache case.
>
>
> ## Reply to Question:
>
> >Will the code be open-sourced? I think it would be valuable for the community.
>
> Yes, upon publication, we will open-source our code to ensure full reproducibility.
>
> Our implementation is based on a widely-used open-source cache compression framework from GitHub. We will also submit a pull request to the official repository to facilitate easy evaluation of our algorithm by the community and to contribute to the broader advancement of the field.
>
>
> ****
>
> **If any concerns remain, please share them—we are happy to discuss further. Should our responses address your concerns, we would greatly appreciate you reconsidering our score. Thank you for your time and thoughtful feedback.**

---

### Official Review · Reviewer_UXkx · 2025-10-31

**Soundness:** 2
**Presentation:** 3
**Contribution:** 3
**Rating:** 6
**Confidence:** 3

**Summary:**

This paper identifies a key issue in KV-cache eviction: prior methods assume that importance scores derived from historical tokens remain stable. However, as generation progresses, these scores can shift significantly, making that assumption unreliable. To address this, the authors propose a worst-case aggregation strategy that accounts for score fluctuation and improves eviction effectiveness.

**Strengths:**

1. This paper identifies a new problem in KV cache eviction methods: the aggregation step ignores the worst-case risk.
2. Comprehensive experiments evaluated the effectiveness of the proposed method.
3. Simple but effective method for optimizing KV eviction methods further.
4. Good writing, easy to follow, and well motivated.

**Weaknesses:**

1. Experiments:

+ The proposed method is orthogonal to many existing KV-cache eviction strategies. To ensure fair comparison and demonstrate the generalizability of your approach, it would be beneficial to include an additional table showing how your method can be combined with representative prior techniques. For example, selecting one model and one benchmark to report results for “baseline methods” versus “baseline + your method” would more clearly highlight the benefits.

      | Methods       | Task 1 | Task2 |
      | ------------- | ------ | ----- |
      | e.g. SnapKV   | ...    | ...   |
      | + DefensiveKV | ...    | ...   |
      | e.g. CAKE     | ...    | ...   |
      | + DefensiveKV | ...    | ...   |

2. Ablation Study:

+ In the default setting, the historical window size is set to 32. According to the paper’s analysis, this hyperparameter plays a crucial role in evaluation, as it affects both scoring and aggregation behaviors. Therefore, it would be valuable to include an ablation study on the historical window size to better understand its impact.

3. Typo:
+ Figure 3 (a) 16st --> 16th

**Questions:**

Please see weaknesses.

---

> ### Author Response · Authors · 2025-11-18
> **Reply to Reviewer  UXkx**
>
> Thank you for your dedicated effort during the review process. We sincerely appreciate your recognition of our work’s conceptual novelty, comprehensive experiments, clear motivation, and simple yet effective technical contributions. We believe these qualities are essential to advancing the efficiency community. Below, we address your remaining concerns in detail.
>
>
>
>
> ## Reply to W1:
> > The proposed method is orthogonal to many existing strategies... an additional table showing combination ...
>
> Yes, our strategy is orthogonal to and can broadly enhance existing eviction methods. In the main paper, we demonstrated the effectiveness when built upon the strongest baseline, CriticalKV. Furthermore, in  Appendix J, we provided additional results when integrated with another baseline, AdaKV.
>
> As suggested, we have consolidated these results into an additional table to more clearly illustrate the compatibility and benefits of our method when combined with prior techniques. The results clearly show that Defensive Aggregation significantly improves the performance of both AdaKV and CriticalKV.
>
> We have incorporated this discussion into the revised manuscript (see Appendix J).
>
> | Llama-3.1-8B, 20% Cache| NrtvQA | Qasper | MF-en | Hotpot | 2WikiQA | Musique | GovRep | QMSum | MultiNews | TREC | TriviaQA | SAMSum | PCount | PR-en | Lcc | RB-P | Avg. |
> |-|-|-|-|-|-|-|-|-|-|-|-|-|-|-|-|-|-|
> | Full Cache | 29.55 | 44.68 | 55.82 | 57.59 | 48.89 | 32.61 | 34.40 | 25.51 | 26.83 | 73.00 | 92.36 | 43.27 | 7.38 | 99.50 | 63.44 | 52.36 | 49.20 |
> | AdaKV | 27.07 | 28.69 | 32.85 | 49.64 | 30.89 | 21.57 | 26.70 | 21.85 | 22.67 | 55.50 | 91.30 | 43.89 | 7.30 | 80.50 | 66.44 | 55.43 | 41.39 |
> | + Defensive | **28.60** | **37.62** | **41.08** | **51.74** | **36.87** | **22.83** | **27.83** | **23.18** | **23.51** | **66.00** | **91.64** | **44.35** | **8.10** | **92.50** | **67.97** | **55.71** | **44.97** |
> |-|-|-|-|-|-|-|-|-|-|-|-|-|-|-|-|-|-|
> | CriticalKV | 29.81 | 32.58 | 34.96 | **52.34** | 36.24 | 26.37 | 28.35 | 23.52 | 23.24 | 56.50 | 90.80 | **43.37** | **8.89** | 93.00 | 67.05 | 54.99 | 43.88 |
> | + Defensive | **29.97** | **40.46** | **46.23** | 52.20 | **38.40** | **28.06** | **29.96** | **23.89** | **24.11** | **68.00** | **91.58** | 43.17 | 8.28 | **100.00**| **67.17** | **55.40** | **46.68** |
>
>
> ## Reply to W2:
> >In the default setting, the historical window size is set to 32...
>
> Thank you for your suggestion. As mentioned earlier, our method extends the strongest baseline, CriticalKV, and we adopted its default window size of 32.
>
> Following your advice, we conducted an ablation study with window sizes of 16 and 64. The results show that DefensiveKV achieves average scores ranging from 46.02 to 46.68, while Layer-DefensiveKV scores between 47.53 and 48.01. Both substantially outperform the strongest baseline, CriticalKV (43.88).
>
> We have added this discussion to the Section 4.4 of the revised manuscript.
>
> | Llama-3.1-8B, 20% Cache | window |  NrtvQA | Qasper | MF-en | Hotpot | 2WikiQA | Musique | GovRep | QMSum | MultiNews | TREC | TriviaQA | SAMSum | PCount | PR-en | Lcc | RB-P | Avg. |
> |-|-|-|-|-|-|-|-|-|-|-|-|-|-|-|-|-|-|-|
> | Full Cache | N/A | 29.55 | 44.68 | 55.82 | 57.59 | 48.89 | 32.61 | 34.40 | 25.51 | 26.83 | 73.00 | 92.36 | 43.27 | 7 | 99.50 | 63.44 | 52.36 | 49.20 |
> | CriticalKV | 32 | 29.81 | 32.58 | 34.96 | 52.34 | 36.24 | 26.37 | 28.35 | 23.52 | 23.24 | 57 | 90.80 | 43.37 | 8.89 | 93.00 | 67.05 | 54.99 | 43.88 |
> |-|-|-|-|-|-|-|-|-|-|-|-|-|-|-|-|-|-|-|
> | DefensiveKV | 16 | 29.75 | 35.43 | **46.54** | **55.51** | 36.41 | 25.64 | 29.20 | 23.68 | 23.34 | 66 | **91.69** | **44.02** | 7.31 | **100.00** | 67 | 55.10 | 46.02 |
> | DefensiveKV | 32 | **29.97** | **40.46** | 46.23 | 52.20 | **38.40** | **28.06** | **29.96** | **23.89** | **24.11** | **68** | 91.58 | 43.17 | **8.28** | **100.00** | **67.17** | **55.40** | **46.68** |
> | DefensiveKV | 64 | 28.98 | 36.09 | 45.49 | 56.69 | 39.31 | 27.50 | **30.63** | 24.12 | 23.31 | **68** | 91.13 | 43.35 | 6.77 | **100.00** | 66.72 | 55.04 | 46.41 |
> |-|-|-|-|-|-|-|-|-|-|-|-|-|-|-|-|-|-|-|
> | Layer-DefensiveKV | 16 | 29.64 | 41.23 | 49.21 | 54.59 | 41.31 | **29.05** | 30.19 | 24.22 | 24.25 | 68.50 | **91.86** | **44.04** | **9** | 99.50 | **68.82** | 55.00 | 47.53 |
> | Layer-DefensiveKV | 32 | 30.10 | 42.91 | **52.94** | 55.03 | **44.07** | 27.00 | 30.99 | **24.95** | **24.42** | 69 | 91.30 | 43.54 | 8.38 | **100.00** | 67.60 | 56.00 | **48.01** |
> | Layer-DefensiveKV | 64 | **30.51** | **43.40** | 51.42 | **55.36** | 40.03 | 26.23 | **31.60** | 24.70 | 24.29 | **70** | 91.61 | 43.21 | 6.92 | **100.00** | 68.69 | **56.46** | 47.75 |
>
> ## W3：Typo
>
> Thank you for pointing this out. We have corrected these typos in the revised manuscript.
>
> *****
>
> **If any concerns remain, please share them—we are happy to discuss further. Should our responses address your concerns, we would greatly appreciate you reconsidering our score. Thank you for your time and thoughtful feedback.**

---

### Author Response · Authors · 2025-12-01
**General Overview**

Dear Reviewers, ACs and SACs,


We thank all reviewers for their feedback and **the initial positive consensus from Reviewers dk2k, WZVm, CK4Z, and UXkx.** Our work is **the first to challenge the long-standing "stability assumption"** in cache eviction, proposing a lightweight defensive aggregation method that reduces compression loss by up to **4.3x**.

We also appreciate **the multiple in-depth discussions with Reviewer Gsr3**, who finally concluded: **“It has addressed my concerns. I will increase my score.”**

We resolved the two main concerns of Reviewer Gsr3 as follows:

1. **Generalization to MoE architectures.** KV cache eviction operates on the attention block and is inherently agnostic to the MoE modifications in the FFN block. We further validated our method on the popular Qwen3-30B-A3B MoE model, confirming its effectiveness.

2. **Performance under extreme compression.**  In the widely adopted realistic evaluation setting with a 20% cache budget, DefensiveKV incurs only 3.9% loss, whereas all baselines exceed 10%—a level generally unusable in practice. As requested, we further report results for a 10% cache budget and fixed budgets of 1024 and 128, where our method consistently and significantly outperforms all baselines.

**Finally, our paper has received unanimous positive recommendations from all five reviewers.** We hope this summary provides a helpful basis for your final decision. Thank you for your consideration.

Sincerely,

Authors

---

### Meta-Review · Area_Chair_rJYu · 2026-01-10

**Summary:**

This paper addresses the KV Cache Eviction problem in LLM inference by identifying and challenging the prevalent "stability assumption" in existing methods. The authors reveal that this assumption is fragile and that Mean Aggregation often leads to degradation in generation quality by overlooking sudden "high-risk" moments. To address this, the paper proposes a "Defensive Aggregation" strategy, which replaces traditional mean aggregation with "Worst-case Risk Estimation" and "Adaptive Prior-Risk Correction." Based on this strategy, the authors introduce DefensiveKV and its layer-adaptive variant, Layer-DefensiveKV. Experiments on benchmarks such as LongBench and Needle-in-a-Haystack demonstrate that these methods significantly outperform SOTA baselines like CriticalKV under a 20% cache budget. The experiments in this paper are extensive and sufficient. Given the strengths mentioned above, I believe this work meets the standards for ICLR. However, I noticed that under the specific settings used in this paper, prior methods consistently suffer from drastic performance drops at a 20% cache budget. This contrasts with empirical observations where methods like SnapKV, CAKE, and AdaKV typically perform well (often remaining near-lossless) under similar settings. I recommend that the authors provide a supplementary comparison or clarification of the method settings to ensure the fairness of the comparison.

**Reviewer Concerns:**

The authors provided a good rebuttal. In the rebuttal, the authors effectively addressed most of the reviewers' concerns, such as compatibility with prior methods and adaptability to MoE models, and effectively answered the reviewers' questions.

**Reviewer Scores:**

Reviewer Gsr3 4->6, while others may maintain their positive ratings.

---

### Decision · Program_Chairs · 2026-01-26

Accept (Poster)